# Cortico-autonomic local arousals and heightened somatosensory arousability during NREMS of mice in neuropathic pain

Romain Cardis[1,2], Sandro Lecci[1], Laura MJ Fernandez[1], Alejandro Osorio-Forero[1], Paul Chu Sin Chung[2], Stephany Fulda[3]*†, Isabelle Decosterd[2]*†, Anita Lüthi[1]*†

[1]Department of Fundamental Neurosciences, Faculty of Biology and Medicine, University of Lausanne, Lausanne, Switzerland; [2]Pain Center, Department of Anesthesiology, Lausanne University Hospital (CHUV), Lausanne, Switzerland; [3]Sleep Medicine Unit, Neurocenter of Southern Switzerland, Civic Hospital (EOC) of Lugano, Lugano, Switzerland

**Abstract** Frequent nightly arousals typical for sleep disorders cause daytime fatigue and present health risks. As such arousals are often short, partial, or occur locally within the brain, reliable characterization in rodent models of sleep disorders and in human patients is challenging. We found that the EEG spectral composition of non-rapid eye movement sleep (NREMS) in healthy mice shows an infraslow (~50 s) interval over which microarousals appear preferentially. NREMS could hence be vulnerable to abnormal arousals on this time scale. Chronic pain is well-known to disrupt sleep. In the spared nerve injury (SNI) mouse model of chronic neuropathic pain, we found more numerous local cortical arousals accompanied by heart rate increases in hindlimb primary somatosensory, but not in prelimbic, cortices, although sleep macroarchitecture appeared unaltered. Closed-loop mechanovibrational stimulation further revealed higher sensory arousability. Chronic pain thus preserved conventional sleep measures but resulted in elevated spontaneous and evoked arousability. We develop a novel moment-to-moment probing of NREMS vulnerability and propose that chronic pain-induced sleep complaints arise from perturbed arousability.

**\*For correspondence:**
stephany.fulda@gmail.com (SF);
isabelle.decosterd@chuv.ch (ID);
anita.luthi@unil.ch (AL)

†These authors contributed equally to this work

## Introduction

Arousability is the capability to wake up from sleep in response to a sensory stimulus. Arousability is a defining behavioral attribute of healthy sleep. Arousal-like events that are brief in time (≤16 s in mouse [*Franken et al., 1999*]; typically 3—15 s in human [*Berry et al., 2018*]) and that occur without an overt sensory stimulus are also part of healthy sleep (*Halász et al., 2004*; *Lo et al., 2004*). The sleeper is often unaware of these brief arousals, although they fragment sleep into bouts of shorter duration. When sleep is excessively fragmented, as is the case in many diverse types of sleep disorders, sleep's restorative effects are severely compromised (*Stepanski et al., 1984*) and long-term health risks increase (*Van Someren et al., 2015*; *Grandner, 2020*). Estimating sleep's vulnerability based on the frequency and types of spontaneous arousals is, therefore, essential to assess the severity of a sleep disruption.

The accurate quantification of spontaneous arousals is indeed a fundamental part in the diagnosis of widespread sleep disorders such as sleep apneas (*Malhotra and Jordan, 2016*). However, this task remains challenging in cases in which arousal origins are complex, such as in insomnias resulting from chronic pain or from psychiatric disorders (*Feige et al., 2013*; *Bjurstrom and Irwin, 2016*; *van*

*Someren, 2020*). Spontaneous arousals seem to occur randomly (*Lo et al., 2004*; *Dvir et al., 2018*), and they can manifest as 'partial' because only some but not all of the physiological correlates of an arousal appear (cortical activation, heart rate increases, muscle tone), as observed in both human (*Bergmann et al., 1987*; *Parrino et al., 2009*) and mice (*Franken, 2002*; *Léna et al., 2004*; *Fulda, 2011*). In humans, arousals additionally occur along a continuum of heart rate changes and cortical activation (*Sforza et al., 2000*; *Azarbarzin et al., 2014*), although such graduation has not been described in rodents. Arousals can occur locally in restricted areas of cortex (*Nobili et al., 2011*; *Castelnovo et al., 2018*). Such variations impede advances in understanding abnormal arousability as a core component of sleep disruptions in both humans and rodent models. In patients, arousals may go undetected and leave sleep fragmentation underestimated. Supporting this possibility, patients complaining about sleep while their polysomnographic measures appear normal remain a subject of intense study (*Bjurstrom and Irwin, 2016*; *Lecci et al., 2020*; *van Someren, 2020*). In rodents, the diversity of spontaneous arousals is poorly studied, which limits the usefulness of animal models of sleep disorders.

The state of non-rapid eye movement sleep (NREMS) is commonly considered a restorative state with an overall low sensory arousability, but also a dynamic state in which various global measures of neuronal excitability and functional connectivity vary on multiple time scales. Of particular interest for arousability is the infraslow time scale that relates to intervals of tens of seconds (*Drew et al., 2008*; *Watson, 2018*). During sleep, the infraslow time scale characterizes variations of brain excitability in epileptic patients (*Vanhatalo et al., 2004*), the clustering of sleep rhythms called sleep spindles in rodents and humans (*Lecci et al., 2017*; *Lázár et al., 2019*), the time course of the human cyclic alternating pattern (*Parrino et al., 2012*), and the dynamics of resting-state networks obtained in functional resonance imaging studies (*Fukunaga et al., 2006*; *Hiltunen et al., 2014*). Infraslow fluctuations in brain excitability and connectivity could render the sleeping brain vulnerable to spontaneous disruptive events, including more frequent arousals in case of sleep disorders.

This study addresses both these aspects in mice. First, we demonstrate in healthy C57Bl/6J mice that infraslow fluctuations determine when spontaneous arousals preferentially occur. Next, we evaluate the case of neuropathic pain, a clinical condition often intractable and of high sensory discomfort in which sleep disruptions are frequent (*Bjurstrom and Irwin, 2016*; *Mathias et al., 2018*). Experimental models of peripheral neuropathic pain rely on surgically crushing or lesioning peripheral nerves, such as the sciatic nerve (*Jaggi et al., 2011*; *Challa, 2015*). In spite of their limitations, these models have allowed numerous breakthrough understandings on chronic pain mechanisms (*Kuner and Kuner, 2020*) and avenues for treatment (*Finnerup et al., 2021*). In contrast, sleep studies on chronic pain models are scarce and have produced variable results (*Andersen and Tufik, 2003*; *Kontinen et al., 2003*; *Tokunaga et al., 2007*; *Cardoso-Cruz et al., 2011*; *Leys et al., 2013*).

We find that the animals with spared nerve injury (SNI) in early stages of chronic pain do sleep normally in terms of sleep macrostructure, although physiological signatures of painful sensations persist in both the waking and the sleeping brain. Using infraslow measures of NREMS vulnerability, in particular the clustering of sleep spindles (*Lecci et al., 2017*), we find that sleep is disrupted in two previously undescribed ways. First, there are more frequent cortico-autonomic local arousals in pain-processing cortical areas while, polysomnographically, sleep remains continuous. Moreover, animals show a higher vulnerability to wake up from NREMS in response to fine vibrational stimuli. The essence of the sleep disturbances in this experimental chronic pain model thus lies in an abnormal arousability, in both its spontaneous and evoked forms.

## Results

### The 0.02 Hz-fluctuation allows to probe variations in spontaneous arousability during NREMS

We first monitored undisturbed sleep in healthy C57Bl/6J mice across the light-dark phase to evaluate the timing of brief arousals in relation to the infraslow fluctuation in sleep spindles. As described previously (*Lecci et al., 2017*; *Yüzgeç et al., 2018*; *Lázár et al., 2019*), these are evident in fluctuations in the sigma frequency range (10–15 Hz) that occur over ~50 s time intervals (~0.02 Hz) and that continue throughout NREMS bouts (*Figure 1A*). Undisturbed NREMS was frequently interrupted by brief arousal events that are called microarousals (MAs), defined in mouse as ≤16 s awakenings

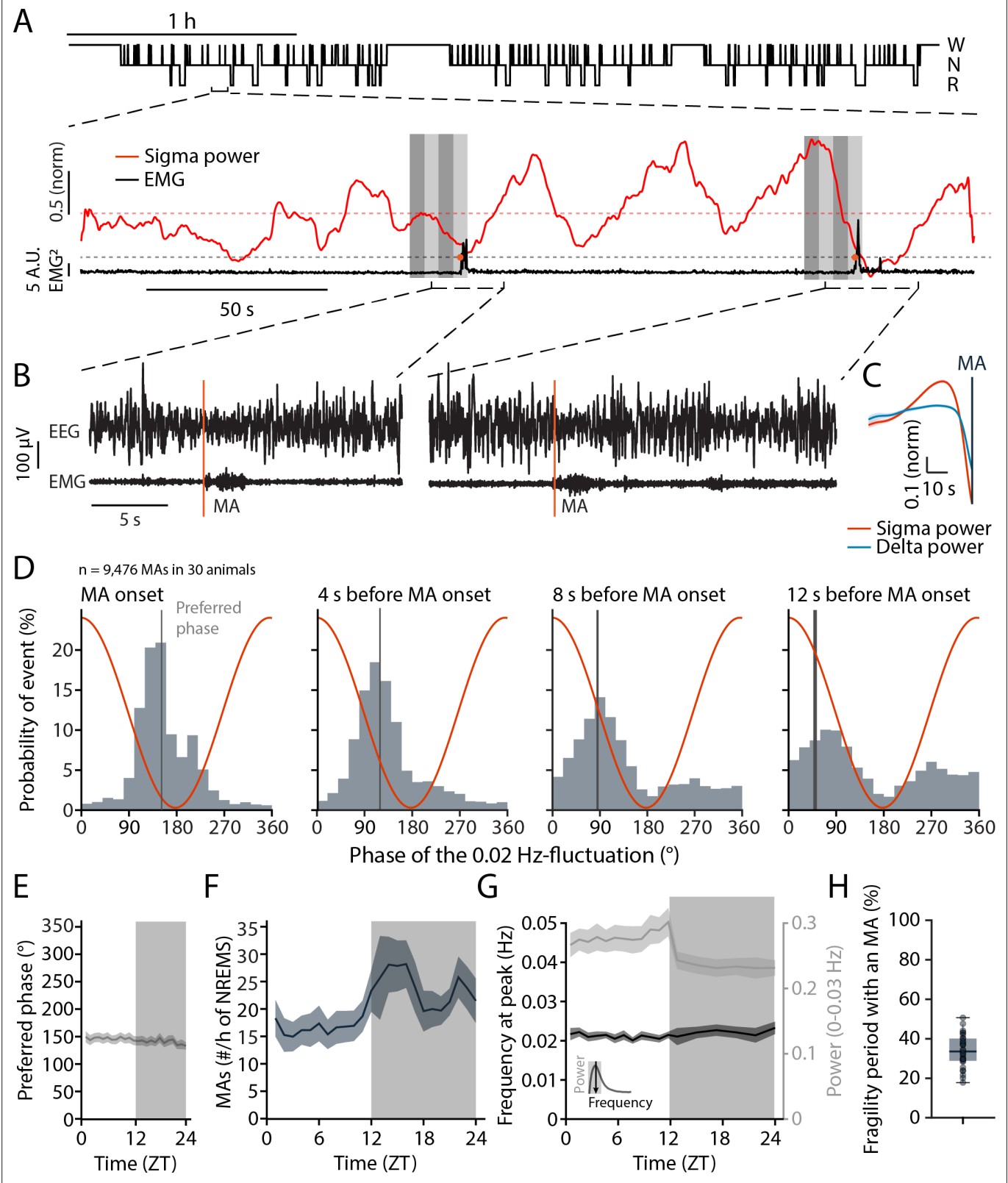

**Figure 1.** Microarousals (MAs) are time-locked to the trough of the 0.02 Hz-fluctuation corresponding to the non-rapid eye movement sleep (NREMS) fragility period. (**A**) Representative mouse hypnogram, W: wakefulness; N: NREMS; R: rapid eye movement sleep (REMS); with an NREMS bout expanded below to indicate the 0.02 Hz-fluctuation in sigma power (red) and the timing of two MAs in the EMG (black). Vertical bars show 4 × 4-s time windows at, and prior to, the onset of, the MAs that were used for phase analysis. (**B**) Two examples of an MA. The MA onset was set at the beginning of

*Figure 1 continued on next page*

*Figure 1 continued*

phasic EMG activity (orange vertical line). (**C**) Mean sigma (10–15 Hz) and delta (1–4 Hz) power dynamics preceding the onset of an MA. (**D**) Histograms of the phase angle values of the 0.02 Hz-fluctuation at specific time points relative to the onset of an MA. The red line represents the corresponding phase of the fluctuation at each bin. (**E**) Preferred phase of the 0.02 Hz-fluctuation at MA onset across time of day in hourly bins (dark phase, shaded, ZT, Zeitgeber time). (**F**) Density of MAs (per hour of NREMS) across the light-dark cycle. (**G**) Parameters of the 0.02 Hz-fluctuation (frequency at peak and power, see inset for illustration) across the light-dark cycle. (**H**) Proportion of fragility periods (corresponding to phase values from 90 to 270°, see panel D) containing an MA. For statistical analysis for this and all subsequent figures, please consult the ***Supplementary file 1***.

The online version of this article includes the following figure supplement(s) for figure 1:

**Source data 1.** Source data for **Figure 1**C–H.

**Figure supplement 1.** Methodological illustrations regarding analysis of the infraslow 0.02 Hz-fluctuation.

accompanied by movement activity seen in the EMG (***Figure 1B***). To evaluate the utility of the 0.02 Hz-fluctuation in sigma power for measures of spontaneous arousability across the entire light phase, we first tested whether MAs were phase-locked to the 0.02 Hz-fluctuation in healthy mice (n = 30 mice with 9476 MAs). The onset of MAs coincided with declining or low sigma power levels that followed a pronounced sigma power peak (***Figure 1B and C***), which is characteristic for a fragility period in which sensory arousability is high (***Lecci et al., 2017***; ***Fernandez and Lüthi, 2020***). A spectral band typical for NREMS, such as delta (1–4 Hz) power, showed a rapid decline preceding the MAs, consistent with EEG desynchronization (***Figure 1C***). The phase values of the 0.02 Hz-fluctuation, calculated via a Hilbert transform (***Figure 1—figure supplement 1***), showed that MA onset times clustered around a mean preferred phase of 151.6° ± 1.1°, with 180° representing the sigma power trough (Rayleigh test, p<1 × 10⁻¹⁶). The majority of MAs (89%) were clustered between 90 and 270°, which narrows the fragility period to the low values of sigma power around the trough (***Lecci et al., 2017***). The phase locking was also observed when time points at 4, 8, and 12 s before the onset of an MA were quantified (***Figure 1D***). This shows that the onset of the fragility period preceded the MA. Fragility periods thus constitute moments during which MAs preferentially occur.

These phase relations persisted for all 1-hr intervals across time of day (***Figure 1E***), although the density of MAs showed a characteristic increase towards the end of the light phase and was higher during the dark phase (***Figure 1F***). The peak frequency of the 0.02 Hz-fluctuation also remained relatively constant, with a minor decrease in power during the dark phase (***Figure 1G***). Across the 24 hr cycle, a median of 33.6% of all fragility periods was accompanied by an MA (***Figure 1H***). In sum, fragility periods were permissive windows for MAs. This means that MAs appeared predominantly during fragility periods, while a majority of fragility periods occurred with NREMS remaining consolidated.

## Three weeks after SNI, mice show signatures of neuropathic pain-related brain activity in wake but normal sleep-wake behavior

Mice with SNI and Sham controls were next analyzed for their sleep-wake behaviors. The time point chosen was at post-surgical days 22–23 (D20+), when chronic pain based on behavioral measures is established (***Decosterd and Woolf, 2000***; ***Bourquin et al., 2006***; ***Guida et al., 2020***). To obtain a correlate of the general behavior of these animals during wakefulness, we implanted Sham and animals with SNI with EEG/EMG electrodes for monitoring of vigilance states and with local field potential (LFP) electrodes. We chose the S1 hindlimb cortex (S1HL, five Sham and nine animals with SNI) (***Figure 2A***), which is the site of sensory discrimination of pain, and the prelimbic cortex (PrL, six Sham and eight animals with SNI) (***Figure 2B***), which is one of the major areas concerned with the affective dimension of pain experience in rodents and in its homologue in humans (***Price, 2000***; ***Moisset and Bouhassira, 2007***; ***Kuner and Kuner, 2020***).

Prior studies have associated elevated power in the theta (5–10 Hz) and gamma (60–80 Hz) frequency bands as pathophysiological correlates of pain states that are sensitive to analgesics or interference with activity in thalamic sensory nuclei in mice or humans (***Sarnthein et al., 2006***; ***LeBlanc et al., 2014***; ***LeBlanc et al., 2016***; ***LeBlanc et al., 2017***; ***Ploner et al., 2017***; ***Tan et al., 2019***). We hence analyzed the spectral composition of S1HL and PrL LFP signals during quiet wakefulness (desynchronized EEG combined with low muscle tone for >12 s) to avoid the confound of exploration-related theta activity (***Figure 2C***), similar to previous studies in mouse (***LeBlanc et al., 2014***; ***LeBlanc***

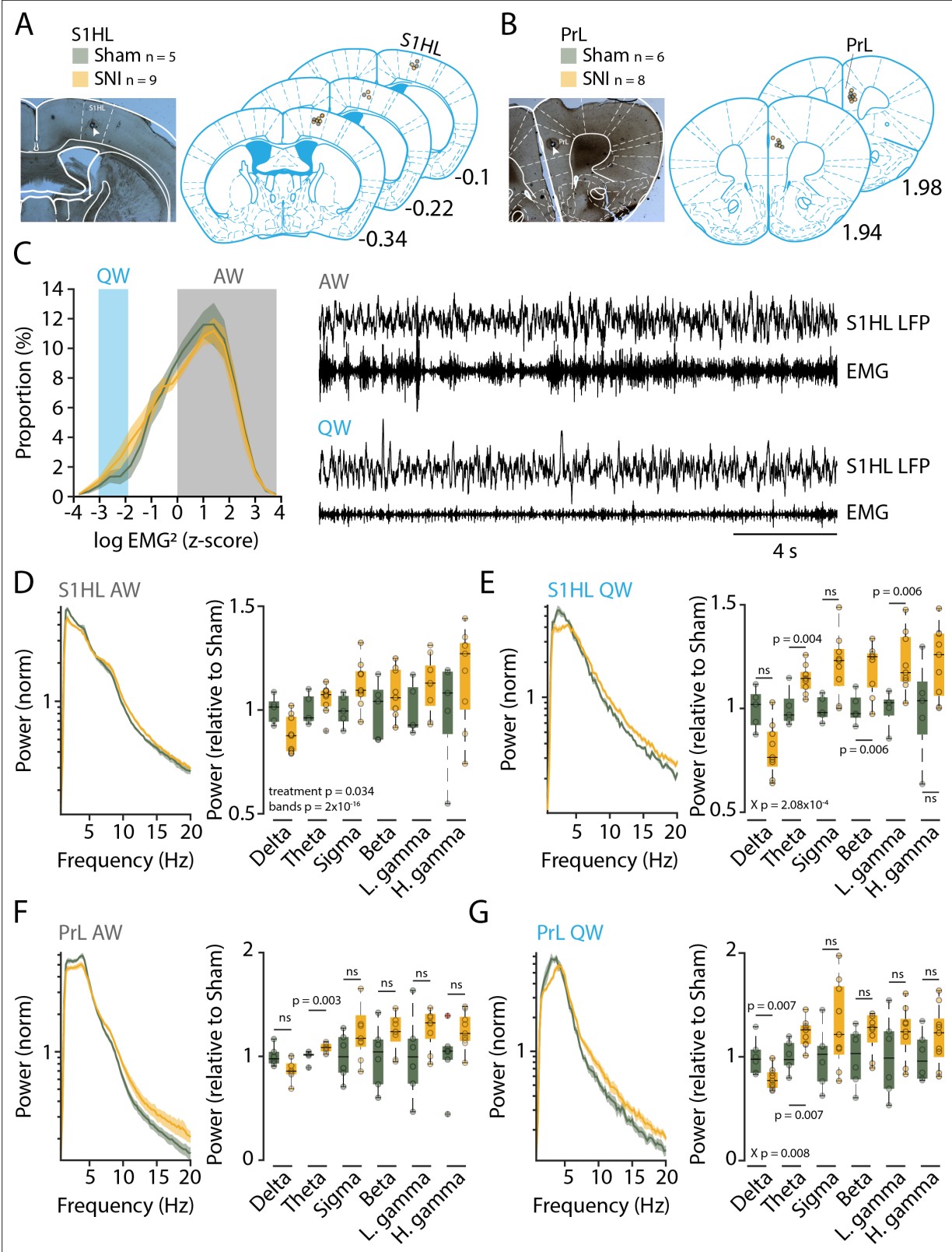

**Figure 2.** Three weeks after spared nerve injury (SNI), animals display alterations in the power spectra of quiet wakefulness within pain matrix areas. (**A, B**) Histological verification of recording sites for S1 hindlimb cortex (S1HL) and prelimbic cortex (PrL). Left panel: representative brain section showing the lesion caused by postmortem electrocoagulation (arrowhead). Right panel: summary of all recording sites across animals (small circles). Anteroposterior stereotaxic coordinates relative to Bregma. (**C**) Histogram showing mean distributions of EMG power values (z-scored) during

*Figure 2 continued*

wakefulness for Sham and SNI. Shaded areas indicate values selected for quiet (QW) and active (AW) wakefulness. Representative traces for QW and AW shown on the right. (**D, E**) Left: log-scaled power spectrum density of S1HL local field potential (LFP) in AW (D) and QW (E) for Sham (n = 5) and SNI (n = 9) animals. Quantification of theta (5–10 Hz), beta (16–25 Hz), and the low gamma (26–40 Hz) frequency range. All data obtained from two successive light phases. For display, the mean level of the Sham for each band was set as 1. Statistics were done on log-transformed values extracted from the spectrum on the left. (**F, G**) Same as (D) and (E) for PrL LFP in AW (F) and QW (G) for Sham (n = 6) and SNI (n = 8) animals. For panel (F), Wilcoxon rank-sum tests were done. In this and all subsequent figures, significant main effects and interactions from the ANOVAs are given. Post-hoc tests were done once interactions were significant, with corresponding p values shown. For (D–G), X p denotes p value obtained from factorial interaction in the mixed-model ANOVAs, p values next to the data points derived from post-hoc tests. Corrected $\alpha = 0.0083$.

The online version of this article includes the following figure supplement(s) for figure 2:

**Source data 1.** Source data for *Figure 2C–G*.

*et al., 2017*). Indeed, power spectral analysis of animals with SNI in quiet but not active wakefulness showed power elevations starting at frequencies around 5 Hz-for S1HL (*Figure 2D and E*, left), leading to increased relative power within the theta (5–10 Hz), beta (16–25 Hz) and the low gamma (26–40 Hz) ranges (*Figure 2E*, right, mixed-model ANOVA for 'treatment' × 'frequency band,' p=2.08 × 10$^{-4}$, post-hoc t-test for SNI vs. Sham, for theta, p=0.004, for beta, p=0.006, for low gamma, p=0.006, with Bonferroni-corrected $\alpha = 0.008$). Similar analyses for PrL indicated a significant increase in the theta band (*Figure 2F and G*, mixed-model ANOVA for G, 'treatment' × 'frequency band,' p=0.008, post-hoc t-test for SNI vs. Sham, for theta, p=0.007, with Bonferroni-corrected $\alpha = 0.008$). We hence concluded that the peripheral nerve injury led to alterations in brain activity that have been described previously as signatures of pain (*LeBlanc et al., 2014*).

An additional group of SNI mice and Sham controls (n = 18 each) were next analyzed for their sleep-wake behavior using standard polysomnography alone. EEG/EMG measurements were carried out prior to SNI and Sham surgery and at post-surgical days 22–23 (D20+) to allow within-animal comparison of sleep-wake behaviors prior to and after establishment of pain hypersensitivity. At both time points, recordings lasted for 48 hr under undisturbed conditions. Based on these data, SNI and Sham controls spent similar amounts of time asleep in the 12 hr light and dark phases, during both baseline and at D20+ (*Figure 3A*). Both treatment groups showed minor increases (2.4–3%) in NREMS time at the expense of wakefulness at D20+ compared to baseline in both light and dark phases (mixed-model ANOVAs with factors 'treatment' and 'day,' p=0.0013 in light phase, p=8.5 × 10$^{-6}$ in dark phase for 'day,' p>0.8 for 'treatment' for either light or dark phase, no interaction). Moreover, cumulative distributions of NREMS and REMS bout lengths at D20+ were similar for Sham and SNI, with only a minor shift toward smaller values in SNI for both NREMS (*Figure 3B*, –2.3 s; Kolmogorov–Smirnov [KS] test, p=0.015) and REMS (–3 s; KS test, p=0.026). When subdivided into short, intermediate, and long bouts for the light phase, there were no significant differences between Sham and SNI for both NREMS and REMS (*Figure 3B*, mixed-model ANOVAs for 'treatment' × 'bout length,' p=0.79 for NREMS, p=0.23 for REMS).

Furthermore, sleep onset latency (*Figure 3C*) and NREMS fragmentation by MAs (*Figure 3D*; *Franken et al., 1999*), were not altered by treatment or time post-surgery (mixed-model ANOVA with factors 'treatment' and 'day,' for sleep onset latency, p=0.42 and p=0.94, no interactions; for number of MAs, p=0.79 and p=0.43, no interactions).

Pain is a body-wide sensation that involves autonomic changes (*Craig, 2003*). We found that heart rate was higher in NREMS after SNI at D20+ compared to baseline (*Figure 3E*, mixed-model ANOVA with factors 'treatment' × 'state' × 'day' with interaction, p=0.02, post-hoc paired t-test for SNI in NREMS, p = 0.002, with Bonferroni-corrected $\alpha = 0.0125$). A tendency was also evident in REMS, during which heart rate is elevated compared to NREMS (*Figure 3E*, effect of 'state' in the ANOVA, p=0.003, paired t-test in SNI in REMS, p=0.027). SNI thus raised heart rate, suggesting elevated sympathetic cardiovascular activation in this model of neuropathic pain.

To examine the presence of SNI-related alterations in the brain rhythms associated with sleep states, we next investigated the mean spectral properties of each vigilance state through constructing normalized power spectral densities (*Vassalli and Franken, 2017*) for the full 48 hr-long recordings. Both NREMS and REMS showed the respective characteristic spectral peaks at delta (1–4 Hz) and at theta frequencies (5–10 Hz), respectively. These were indistinguishable between the two groups of animals and from baseline to D20+ (*Figure 3F and G*). In contrast to the preserved low-frequency

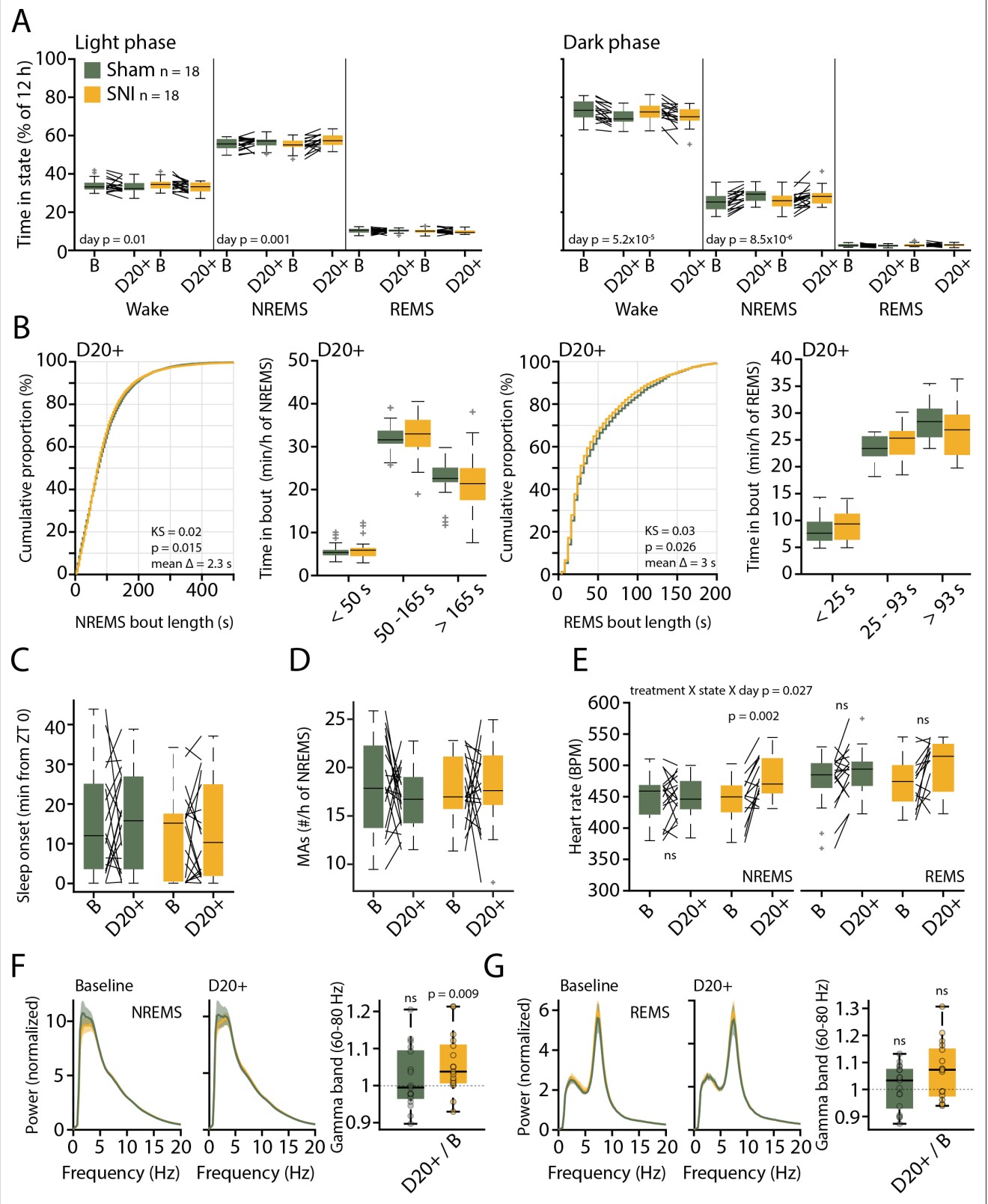

**Figure 3.** Preserved sleep-wake behavior accompanied by pathological changes in sleep after spared nerve injury (SNI). (**A**) Mean percentage of total time spent in the three main vigilance states for Sham (n = 18) and SNI (n = 18) animals in light (left) and dark phase (right) in baseline (B) and at day 20+ (D20+) after surgery. Black lines connect data from single animals. Mixed-model ANOVAS were applied. (**B**) Bout size cumulative distribution (with Kolmogorov–Smirnov [KS] test results) and time spent in short, intermediate, and long bouts for non-rapid eye movement sleep (NREMS) (left) and rapid

*Figure 3 continued*

eye movement sleep (REMS) (right) between Sham and SNI at D20+. Mixed-model ANOVA for time in bouts (a log transform for normality criteria was used for NREMS). (**C**) Mean latency to sleep onset. Mixed-model ANOVA, not significant (ns). (**D**) Number of microarousals (MAs) per hour of NREMS in light phase. Mixed-model ANOVA, ns. (**E**) Heart rate in NREMS (left) and REMS (right) from animals with suitable EMG signal, Sham (n = 17) and SNI (n = 14). Mixed-model ANOVA with three factors interaction followed by post-hoc tests. Corrected α = 0.0125. (**F**) Left: normalized power spectrum for Sham and SNI for NREMS at baseline and D20+. Shaded errors are 95% confidence intervals of the means. Right: high-gamma power (60–80 Hz) for D20+ relative to baseline (from the spectra shown on the left). One-sample t-tests with corrected α = 0.0167. (**G**) Same display as (F) for REMS. Spectra are presented up to 20 Hz to highlight the characteristic peaks for NREMS and REMS at low frequencies.

The online version of this article includes the following figure supplement(s) for figure 3:

**Source data 1.** Source data for *Figure 3A–G*.

power, relative gamma power was increased in SNI at D20+ compared to baseline in NREMS (*Figure 3F*, one-sample t-test for NREMS in SNI, p=0.0092). There were no correlations between relative changes in gamma power and alterations in sleep architecture in individual mice (change in the number of MAs per hour of NREMS × change in gamma power; pairwise linear correlation $R^2 = 0.09$, p=0.08; change in total NREMS time × change in gamma power; $R^2 = 0.02$, p = 0.36).

These data indicate that after SNI mice do not suffer from major alterations in sleep-wake behaviors. However, pathological increases in wake-related brain rhythms and increases in heart rate continued to be present in sleep. We next tested whether such alterations generate spontaneous arousals that were timed to the infraslow fluctuation.

## NREMS after SNI shows normal phase-coupling of MAs to the 0.02 Hz-fluctuation

The 0.02 Hz-fluctuation was not different between Sham and SNI (n = 18 for both groups) across the light phase. Thus, neither its amplitude nor frequency (*Figure 4A–C*), or, equivalently, the number of its cycles per hour of NREMS, were different between the groups (*Figure 4D*). The phase-coupling to MAs was also unaltered (*Figure 4E*, mean angle ± 95% confidence interval [CI]: 152.3 ± 1.4 for Sham and 150.4 ± 1.3 for SNI) and the distribution of fragility periods containing transitions to MAs, to REMS, or with continuation into NREMS was indistinguishable (*Figure 4F*).

It has been shown that sleep loss exacerbates pain (*Alexandre et al., 2017*). Sleep could thus be relatively more disrupted in the SNI group after a period of sleep loss. We therefore carried out a 6 hr sleep deprivation (SD) at the beginning of the light phase as done previously in the lab (n = 12 for Sham and SNI each) (*Kopp et al., 2006*). We confirmed a characteristic rebound of delta power (*Figure 4G and H*) and a decrease in the frequency of MAs (*Figure 4I and J*, mixed-model ANOVA with factors 'treatment' and 'SD,' p=0.35 and p=1.23 × 10⁻⁷ with no interaction). The phase-coupling of MAs to the 0.02 Hz-fluctuation remained unaltered in both groups even with high sleep pressure (*Figure 4K and L*). SNI thus left spontaneous MAs, their coupling to the 0.02 Hz-fluctuation, as well as homeostatic regulation of spontaneous arousability unaltered.

## NREMS after SNI shows more frequent cortical local arousals

We next tested whether the 0.02 Hz-fluctuation could serve to identify previously undescribed arousal types in mice with characteristics distinct from conventional MAs. For this, we generated spectral profiles of all cycles of the 0.02 Hz-fluctuation that were devoid of MAs and continued into NREMS. We used the data obtained from stereotaxically guided LFP recordings (see *Figure 2*) to ensure that local arousal events in pain-related cortical areas did not escape detection (*Fernandez et al., 2018*). LFP recordings reliably reported on the 0.02 Hz-fluctuation in S1HL and PrL. Consistent with its predominant expression in sensory cortices (*Lecci et al., 2017*), the 0.02 Hz-fluctuation showed a higher peak in S1HL than in PrL (*Figure 5A and B*). The cycles of successive continuity and fragility periods were extracted (see *Figure 1—figure supplement 1*) and their spectral dynamics plotted separately for the relative contribution of power in the low-frequency delta (1–4 Hz) and the beta (16–25 Hz), low- (26–40 Hz), and high- (60–80 Hz) gamma bands (*Figure 5C–F* for S1HL, *Figure 5G–J* for PrL). Average values for the infraslow phase angles between 90 and 270°, corresponding to the fragility period enriched in MAs (see *Figure 1*), and for the continuity period (from 270 to 90°), were calculated. Consistent with the selection of fragility periods continuing into NREMS, delta power remained high and even increased further during fragility periods in S1HL (*Figure 5C*). Prior studies

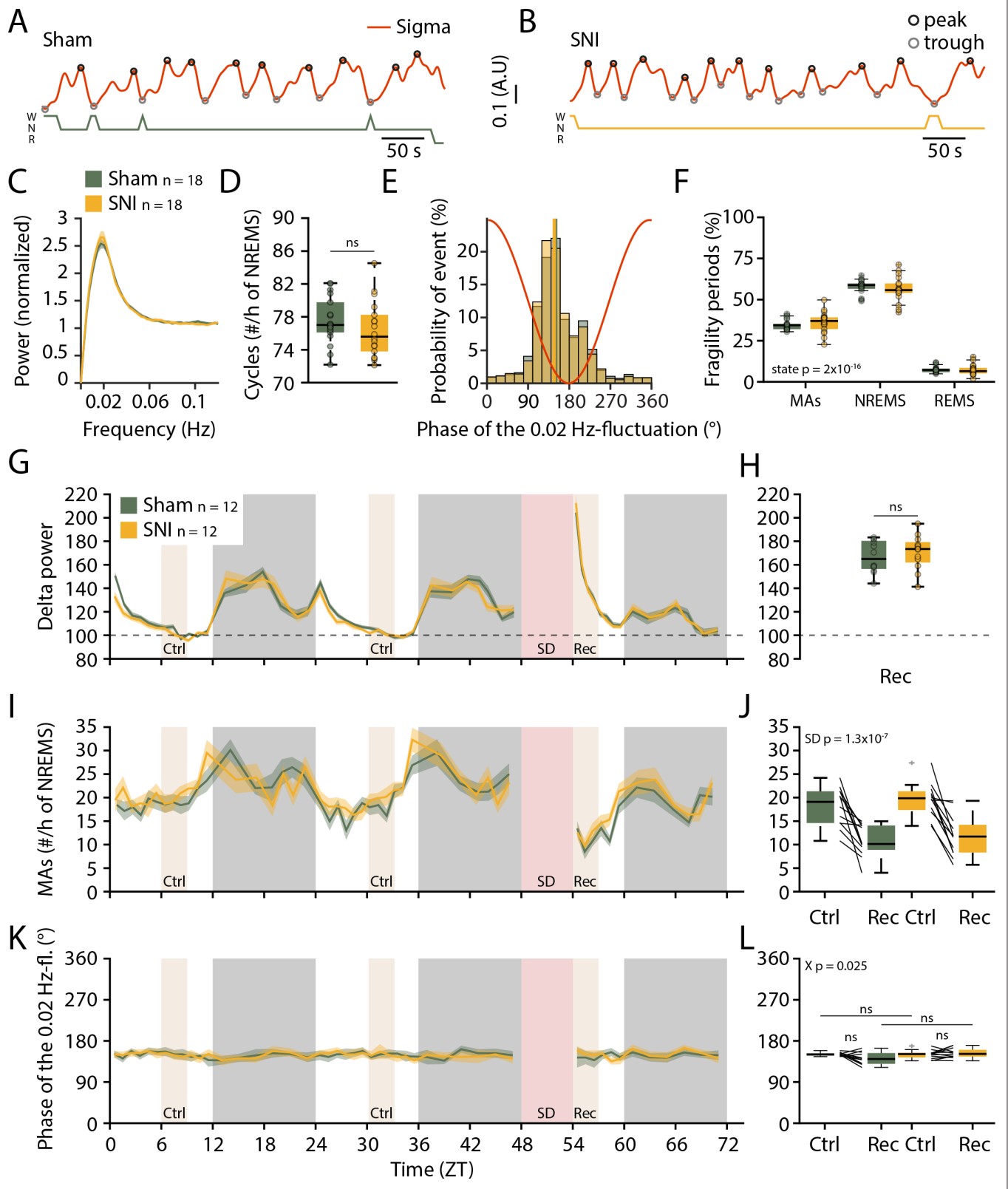

**Figure 4.** Preserved 0.02 Hz-fluctuation and phase-coupling to microarousals (MAs) three weeks after spared nerve injury (SNI). (**A, B**) Representative traces of sigma power dynamics, with hypnograms shown below for W: wakefulness; N: non-rapid eye movement sleep (NREMS); R: rapid eye movement sleep (REMS). The circles represent the result of detection of individual cycles of the fluctuation, as described in Materials and methods. (**C**) Power in the infraslow range extracted from sigma power dynamics for Sham and SNI (n = 18 each); shaded areas represent 95% confidence intervals

*Figure 4 continued on next page*

*Figure 4 continued*

(CIs). (**D**) Number of infraslow cycles per hour of NREMS. Data are shown for D20+ only, but baseline data points were included in the statistical analysis. (**E**) Histograms of the phase values of the 0.02 Hz-fluctuation at MA onset for Sham and SNI, calculated as in *Figure 1D*. Vertical lines denote mean phase ± 95% CI for Sham: 152.3 ± 1.4; SNI: 150.4 ± 1.3. (**F**) Proportion of fragility periods (defined by 0.02 Hz-fluctuation phase values of 90–270°) containing an MA, continuing into NREMS or containing a transition to REMS. (**G**) Delta power dynamics across two light and dark phases and after a 6 hr sleep deprivation (SD). Rec: recovery period; Ctrl: control periods. Delta power values are normalized to the mean of those at ZT9-12. Shaded areas represent SEM. SD was carried out on a subset of 12 Sham and 12 animals with SNI on the day following the D20+ recording. (**H**) Boxplot for delta power values during Rec, corrected α = 0.0125. (**I, J**) As (G, H) for the number of MAs per hour of NREMS. (**K, L**) As (G, H), for the preferred phase of the 0.02 Hz-fluctuation at MA onset. In this figure, mixed-model ANOVAs followed by parametric post-hoc tests or nonparametric tests were used with corrected α = 0.0125.

The online version of this article includes the following figure supplement(s) for figure 4:

**Source data 1.** Source data for *Figure 4C–L*.

showed that in rodent sensory cortices local sigma and delta power dynamics run opposite to each other during NREMS, in part because thalamic mechanisms underlying these local rhythms are shared (*Fernandez et al., 2018*; *Fernandez and Lüthi, 2020*). Analysis across power bands revealed that, after SNI, mice showed alterations in relative amplitude and dynamics in S1HL, but not in PrL. In particular, delta power was lower after SNI during fragility periods (*Figure 5C*, mixed-model ANOVA with factors 'treatment' and 'period,' p=0.001 for the interaction). While in Sham controls there was a distinct rapid upstroke of power in this frequency band that reached a peak during the fragility periods (post-hoc t-test for delta power in fragility vs. continuity period in Sham, p=0.001), there was no such elevation detectable in delta power in SNI (post-hoc t-test in SNI, p=0.44). This difference indicates that fragility periods after SNI were destabilized compared to Sham. Further supporting this, the high-frequency bands in the beta and low-gamma range showed a tonic increase after SNI that was present throughout continuity and fragility periods in S1HL (*Figure 5D–F*, mixed-model ANOVA with factors 'treatment' and 'period,' for beta, p=0.01, p=$1.3 \times 10^{-9}$ and for low gamma, p=0.005, p=$1.4 \times 10^{-10}$) and that was also present, although to a milder extent, in the high-gamma range (*Figure 5F*).

We calculated an 'activation index' (AI), defined by the ratio between high- and low-frequency spectral power components (more precisely, the ratio between summed spectral power in the beta and low-gamma bands over the delta band power). This AI is a measure for the degree of arousal and increases when NREMS moves closer to wakefulness (*Lecci et al., 2020*). In the fragility periods continuing into NREMS and being devoid of EMG activity, the AI decreased, consistent with NREMS remaining consolidated (*Figure 6A–C*). After SNI, however, the mean AI was higher compared to Sham specifically in the fragility periods (*Figure 6B*, mixed-model ANOVA with factors 'treatment' and 'period,' p=0.039 for interaction, post-hoc t-tests Sham vs. SNI in fragility period, p=0.005, in continuity period, p=0.027, not significant with α = 0.0125). Fragility periods during uninterrupted NREMS were thus moments during which the AI after SNI was specifically higher compared to continuity periods.

Can such mean differences in local cortical activation profiles during NREMS qualify as cortical arousals? To address this, we compared the AI in MA-free fragility periods with the AI in an MA that is associated with EMG increase and strong cortical activation. As expected, the AI showed an intermittent phasic peak (*Figure 6D–F*) in most MAs (75.2% ± 4.1%), which is explained by the strong decline in delta power (see *Figure 1C*) and the appearance of higher frequencies associated with MAs. Therefore, we inspected individual fragility periods continuing into NREMS (without an MA) for the presence of similar phasic increases in AI. Indeed, we noticed that a subset of these did contain an intermittent peak resembling the one found during MAs (*Figure 6G*) and not evident in the mean AI in *Figure 6B*. These events could qualify as a local cortical arousal based on phasic spectral properties reminiscent of an MA (*Figure 6—figure supplement 1A-C*). To further support our assumption that these AI peaks constituted arousals, we looked at heart rate increases known to accompany cortical arousals in human (*Sforza et al., 2000*; *Azarbarzin et al., 2014*). The heart rate was distinctly higher during the fragility period for cycles containing an AI peak as opposed to the ones without such peak (*Figure 6H*, mixed-model ANOVA with factors 'treatment,' 'period,' and 'peak', p=0.007 for the 'peak' × 'period' interaction). Although they are present in both groups, these events were more frequent after SNI and followed a similar time of day dependence as the classical MAs ((*Figure 6I and J*), t-test Sham vs. SNI, p=0.02). Moreover, their increased occurrence was specific for S1HL

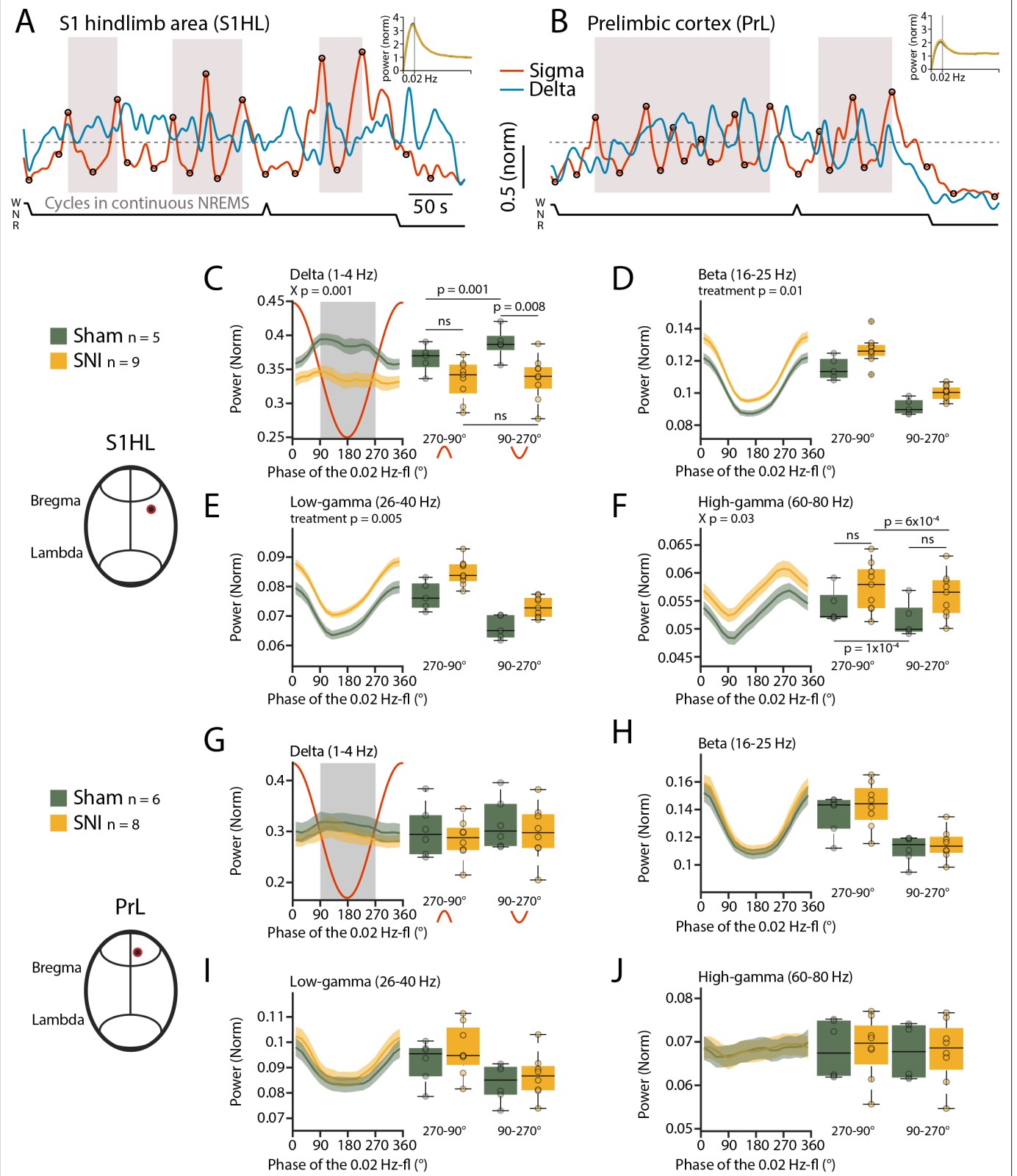

**Figure 5.** Spared nerve injury (SNI) induces locally disrupted spectral power dynamics during non-rapid eye movement sleep (NREMS). (**A, B**) Sigma (10–15 Hz) and delta (1–4 Hz) dynamics during the same NREMS period in S1 hindlimb cortex (S1HL) (**A**) and prelimbic cortex (PrL) (**B**) during a baseline recording. Hypnograms shown below. Circles represent peaks and troughs used to detect the 0.02 Hz-fluctuation cycles (see Materials and methods). Shaded areas indicate 0.02 Hz-fluctuation cycles that are continuing uninterrupted in NREMS. Insets: power spectra of sigma power dynamics in the

*Figure 5 continued on next page*

*Figure 5 continued*

infraslow range. (**C–F**) For S1HL, trajectories of power in specific frequency bands across uninterrupted cycles of the 0.02 Hz-fluctuation, quantified in 20° bins. Red line shows the corresponding 0.02 Hz-fluctuation phase. Boxplots quantify spectral power within continuity (270–90°, red inverted U-shaped line) and fragility periods (90–270°, red U-shaped line). For all bands, mixed-model ANOVAs were done, followed by post-hoc t-tests if applicable. Significance for main effects and/or interaction (X p) is shown on top of the graphs, and post-hoc tests with corrected α = 0.0125 are presented next to the data. (**G–J**) Analogous presentation for PrL.

The online version of this article includes the following figure supplement(s) for figure 5:

**Source data 1.** Source data for *Figure 5A–J*.

while absent in PrL and in the contralateral EEG (*Figure 6—figure supplement 1*). The presence of a subgroup of fragility periods continuing into NREMS, yet showing a cortical arousal is noteworthy for several reasons. First, it demonstrates that rodent NREMS shows local cortical intrusion of wake-related activity in the absence of muscular activity. Second, these local cortical arousals in SNI showed intermittent peaks in AI that were close to the ones of MAs, indicating comparable cortical desynchronization at the local level. Third, they were accompanied by heart rate increases that are sensitive hallmarks of arousal in human (*Azarbarzin et al., 2014*). Fourth, nerve injury increased the relative occurrence of fragility periods with such AI peaks specifically in the S1HL area. The systematic classification of fragility periods helped unravel these novel cortico-autonomic arousals and their similarity to MAs. Still, other arousal-like events outside fragility periods could exist.

## The 0.02 Hz-fluctuation allows to anticipate elevated spontaneous arousability during NREMS

We finally examined sensory arousability 3 weeks after SNI, focusing on the somatosensory modality. To anticipate fragility and continuity periods in real time in the sleeping animal, we trained a machine learning software to predict online periods of continuity and fragility based on EEG/EMG recordings (*Figure 7A–E*). For the training, we used online-calculated 0.02 Hz-fluctuation estimates for which fragility and continuity periods were labeld using peak-and-trough detection of sigma power dynamics (*Figure 7—figure supplement 1*). To control for the accuracy of the online prediction, we visually scored MAs in 12 C57Bl/6J animals implanted only for polysomnography and verified their position in either online-detected peak-to-trough ('online fragility') or trough-to-peak phases ('online continuity') (*Figure 7F*). We compared the online prediction to that generated by chance through randomly shuffling both online fragility and continuity point positions in the recordings. This showed that the MA proportions obtained with the real detection exceeded those obtained by chance prediction (*Figure 7G*, for online fragility periods, p=0.0004, for online continuity periods, p=0.0028). Online detection of peak-to-trough and trough-to-peak periods of the 0.02 Hz-fluctuation is thus a versatile method to probe variations of evoked arousability from NREMS.

## SNI produces elevated sensory arousability from NREMS

Evoked arousability was probed through applying mechanical stimuli on the skull during either online-detected fragility or continuity periods. To deliver somatosensory stimuli remotely while the animals were asleep, we attached vibrational motors to their head implant that could be triggered to briefly (3 s) vibrate and gently shake head and body portions, to test the chance for wake-up (*Figure 8A*). These motors were calibrated to vibrate with the same low intensity (~30% of full power) across animals (*Figure 8—figure supplement 1*). Vibrations were applied randomly with 25% probability during online-detected fragility or continuity periods for at least two complete light phases per condition (*Figure 8B*). Intensity was chosen such that after Sham surgery mice showed approximately equal chances for wake-up or sleep-through in online continuity periods (*Figure 8C*). Moreover, these vibrations produced wake-ups that were short, suggesting that the sleeping animal seems only mildly perturbed. Consistent with prior findings, similar stimuli applied during online fragility periods showed consistently higher chances for wake-up (*Lecci et al., 2017*). After SNI, sensory arousability was elevated for both continuity and fragility periods, leading to highest values during the online fragility periods (*Figure 8D*, mixed-model ANOVA with factors 'treatment' and 'online period,' p=0.0049 for 'treatment,' p=1.31 × 10$^{-8}$ for 'online period,' no interaction). Interestingly, consistent with the tonic

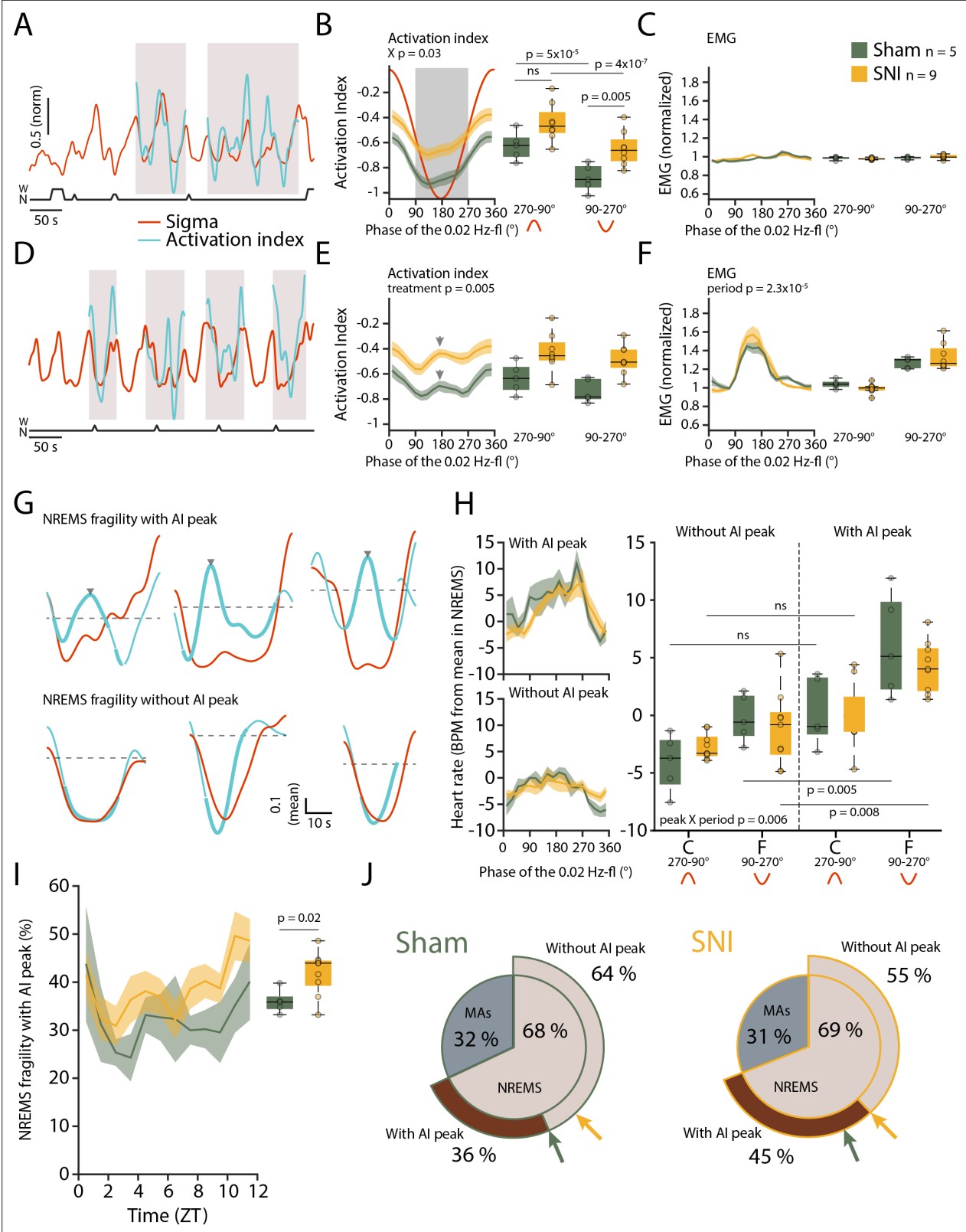

**Figure 6.** Mice generate local cortical arousals during non-rapid eye movement sleep (NREMS) that appear more frequently 3 weeks after spared nerve injury (SNI). (**A**) Normalized dynamics of sigma power (10–15 Hz) and activation indices (calculated as the natural logarithm of (beta + low-gamma)/delta). Hypnogram is shown below. Shaded areas show cycles of the 0.02 Hz-fluctuation continuing into NREMS. (**B**) Activation indices (AIs) for cycles continuing into NREMS, with mean values shown in boxplots for continuity (270–90°) and fragility periods (90–270°). Mixed-model ANOVAs followed

*Figure 6 continued on next page*

*Figure 6 continued*

by post-hoc t-tests, corrected α = 0.0125. (**C**) Corresponding EMG values (normalized to the mean value in NREMS). Mixed-model ANOVA, ns: not significant. (**D–F**) Same as (A–C), for cycles of the 0.02 Hz-fluctuation associated with microarousals (MAs). Shaded areas in panel (D) show cycles of the 0.02 Hz-fluctuation interrupted by an MA. Arrowhead in (E) denotes the peak of the intermittent increase in AI due to MA occurrence. Mixed-model ANOVAs yielded single-factor effects. (**G**) Six individual cases in one Sham animal illustrating sigma (red) and AI (blue) dynamics in uninterrupted cycles of the 0.02 Hz-fluctuation. Thick portions of the blue traces represent the AI during the fragility period. Top and bottom three examples show an AI without and with an intermittent peak, respectively. The horizontal line (mean AI per cycle) represents the threshold for peak detection. (**H**) Left: heart rate dynamics for cycles continuing into NREMS. Cycles are divided into whether or not an AI peak was present. Right: boxplot quantification for continuity (C, 270–90°) and fragility periods (F, 90–270°), for Sham and SNI. Three-factor mixed-model ANOVAs followed by post-hoc t-tests, corrected α = 0.0125. (**I**) Occurrence of fragility periods with intermittent peak in AI across the light phase. Unpaired t-test. (**J**) Two level-pie plots for Sham (left) and SNI (right) representing the proportion of cycles containing an MA or continuing into NREMS. These latter fragility periods are further subdivided into the ones with intermittent peak ('peak') and without intermittent peak ('no peak'). The arrows show the proportions for Sham (green) and SNI (yellow).

The online version of this article includes the following source data and figure supplement(s) for figure 6:

**Source data 1.** Source data for *Figure 6B,C,E,F,H,I*.

**Figure supplement 1.** Activation index (AI) with peaks in fragility period in EEG and prelimbic cortex (PrL) local field potential (LFP) recordings.

**Figure supplement 1—source data 1.** Source data for Figure 6—figure supplement 1B–F.

increase in AI, this increase in sensory arousability after SNI was present across the whole light phase with a conserved time of day dependence (*Figure 8E*).

## Discussion

Chronic pain is a widespread and complex condition compromising sleep. As poor sleep further aggravates pain, and sleep disturbances impact quality of life, therapeutic approaches to target the sites of interaction between sleep and pain are of particular interest. Here, in efforts to tease apart pain-sleep associations, we focused on abnormal arousability at moments when NREMS is most vulnerable. This study progresses on the sleep-pain association in four essential ways. First, we show that brain and autonomic signatures of SNI during the day intrude in a persistent manner into sleep. Second, sleep appeared nevertheless preserved in architecture, dominant spectral band power, and homeostatic regulation. Third, a previously undescribed spontaneous arousal during NREMS, showing local cortical activation with concomitant heart rate increases, appeared more frequently after SNI. Fourth, we also demonstrate that mechanovibrational stimuli of the body triggered brief wake-ups from NREMS more easily in the SNI group of mice. In summary, an experimental model of chronic pain impacts on NREMS in terms of arousability, more specifically on the probability that NREMS transits towards diverse levels of wakefulness, either spontaneously, or with external stimuli.

Accumulating evidence indicates that infraslow time scales relate to variations in sensory arousability (*Lecci et al., 2017*; *Yüzgeç et al., 2018*). We found here that spontaneous MAs without obvious sensory input occur clustered at moments when sensory arousability is high (*Lecci et al., 2017*). Therefore, when mouse NREMS is vulnerable to sensory stimuli, it is also vulnerable to spontaneous arousals. Sensory arousals and MAs may have common mechanistic origins, in line with evidence that both depend on activity in wake-promoting brain areas (*Léna et al., 2004*; *Huang et al., 2006*; *Dvir et al., 2018*; *Mátyás et al., 2018*). Furthermore, the result suggests that infraslow time scales determine the vulnerability of NREMS for both weak (in case of spontaneous arousals) and more intense (in case of evoked arousals) levels of wake-promoting input. The increased delta power and decreased AI in S1HL in fragility periods continuing into NREMS seem at odds with this finding because they signal momentarily increased sleep depth. These dynamics are likely a consequence of an incompatibility between sigma and delta power in sensory thalamocortical loops (*Fernandez et al., 2018*) that are not, however, strong enough to override the vulnerability of NREMS identified here. Intriguingly similar dynamics are also observed for CAP, in which periodic increases in delta alone are part of the arousal pattern (*Parrino et al., 2012*). This motivates studies testing the usefulness of fragility periods as time raster in the search for diverse, even subtle, arousal-like events that could be relevant to model pathological conditions of human patients.

The major portion of this study aimed to obtain a proof of concept for this possibility. The SNI model appeared particularly appropriate because it presents a specific pain hypersensitivity that could be investigated with respect to its impact on both, abnormal spontaneous and evoked arousability,

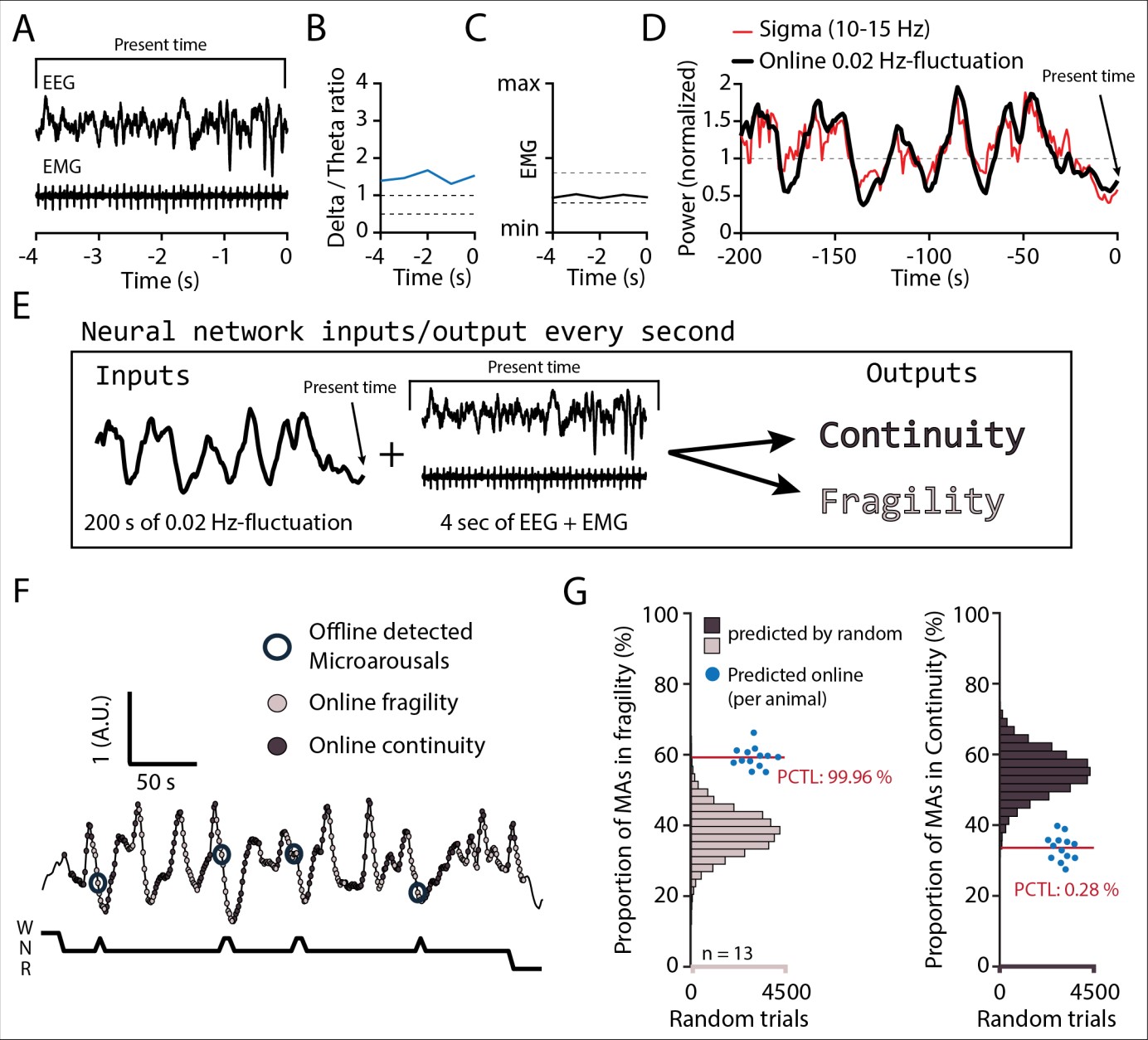

**Figure 7.** Online detection of fragility periods during mouse non-rapid eye movement sleep (NREMS). (**A–E**) Input and output parameters for machine learning of NREMS fragility and continuity periods. The network was trained to use the last 200 s of online 0.02 Hz-fluctuation and the present window of EEG and EMG values to determine whether the animal was in continuity or in fragility periods. (**F**) Representative result of an online detection of fragility (light gray circle) and continuity (dark gray circle) periods. The hypnogram below represents the visual scoring done offline, with detected microarousals (MAs) superimposed by open circles over the corresponding point of online detection. (**G**) Proportion of MAs scored during online-detected fragility (left) or continuity (right) for 13 animals (blue dots). Horizontal histograms represent the distribution of possible values of these proportions for randomly shuffled points of fragility or continuity. The mean proportions for the 13 animals fell at percentile (PCTL) 99.96% for fragility and 0.28% for continuity.

The online version of this article includes the following figure supplement(s) for figure 7:

**Source data 1.** Source data for *Figure 7G*.

**Figure supplement 1.** Online 0.02 Hz-fluctuation extraction and training of the neural network.

during NREMS. The availability of fragility periods as time raster guided the identification of arousals that occurred locally while NREMS continued uninterrupted and that were more pronounced and more frequent in SNI. Without the raster provided by the fragility periods, the phasic differences in AI amidst the tonically elevated high-frequency power would easily have gone undetected. To further

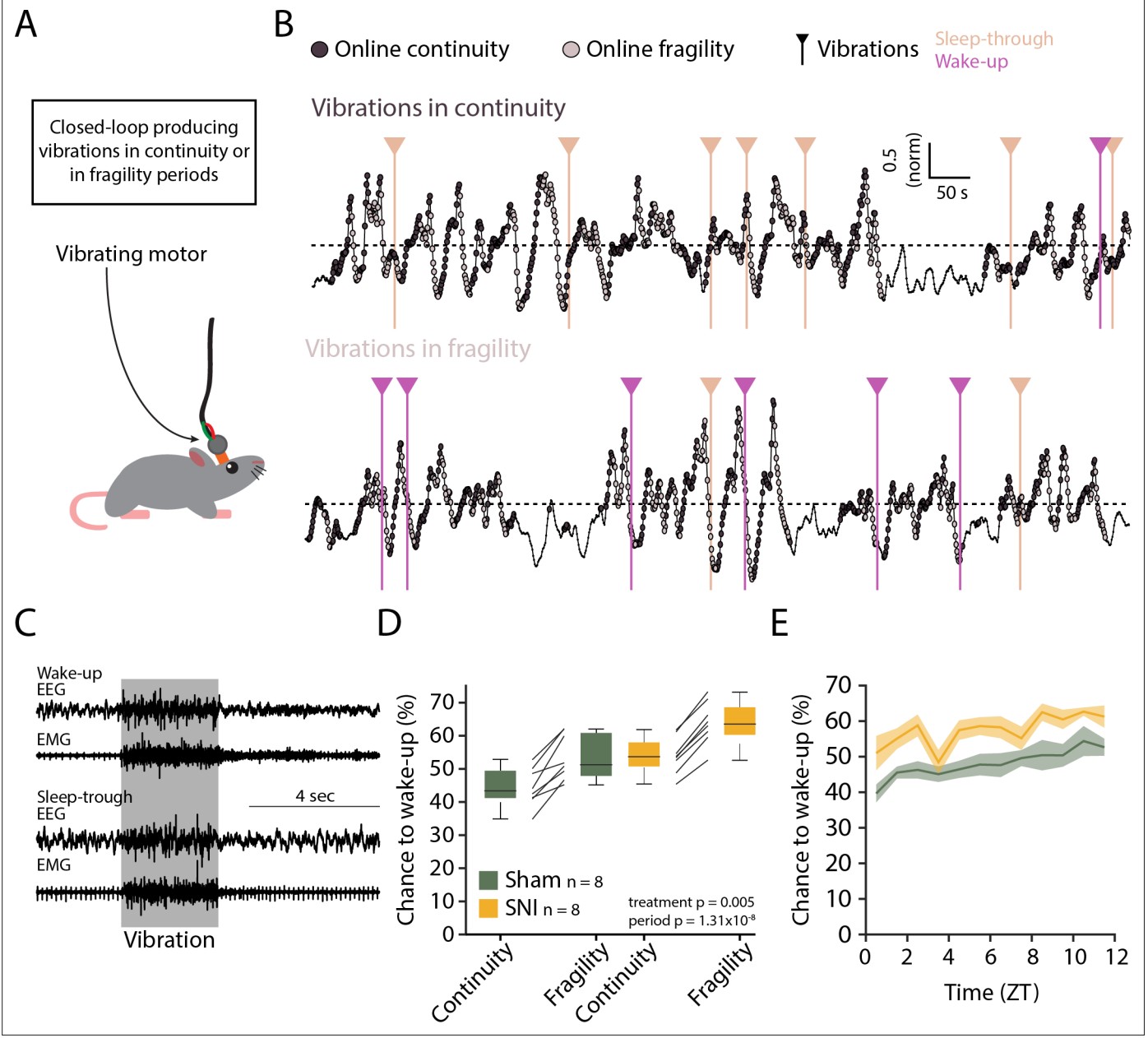

**Figure 8.** Spared nerve injury (SNI) induces higher somatosensory arousability. (**A**) Experimental set-up. A small vibrating motor was fixed at the end of the recording cable that was connected to the animal's headstage. Using the algorithm presented in *Figure 7*, a 3-s mild vibration (see Materials and methods) was delivered with a chance of 25% during either continuity or fragility periods. (**B**) Example experiments. Dark and light circles represent the online detection. Vertical lines represent the moments when the vibrations were delivered. Color codes for wake-ups or sleep-throughs. (**C**) Representative EEG/EMG recordings of a wake-up (top) and a sleep-through (bottom). Time of stimulus delivery is shaded. (**D**) Chances to wake-up in response to a vibration in online continuity or online fragility periods. A mixed-model ANOVA was done, yielding significant factor effects but no interaction. (**E**) Probability of wake-up (pooled continuity and fragility) for Sham and SNI over the light phase (ZT0-12). Shaded areas are SEMs.

The online version of this article includes the following figure supplement(s) for figure 8:

**Source data 1.** Source data for *Figure 8D, E*.

**Figure supplement 1.** Vibration motor calibration for closed-loop somatosensory arousability testing.

ascertain that these detected events constituted true arousals rather than accidental spectral fluctuations, we sought for independent physiological correlates. Heart rate increases were consistently higher when calculated for fragility periods with AI peak than for the ones without AI peak. Moreover, their distribution across the resting phase was similar to the one found for MAs and they were

present in both S1HL and PrL. This result supports our interpretation that we have identified here a novel cortical-autonomic arousal subject to similar time-of-day-dependent regulatory mechanisms as conventional MAs. It also concurs with human NREMS in which heart rate increases occur even for very weak cortical arousals (*Azarbarzin et al., 2014*). Still, we cannot exclude that arousal subtypes outside the fragility periods went undetected that would require further characterization. We also remark here clearly that, aside from more frequent cortico-autonomic arousals, the SNI group of mice suffered from a tonically elevated high-frequency power in S1HL that likely underlay the more elevated sensory arousability throughout continuity and fragility periods.

We found that the cortical regions in which arousals appeared coincided with sites known to be affected by neuropathic pain. The continued presence of high-frequency activity in S1HL is reminiscent of the cortical oscillatory activity present during withdrawal from acute painful stimuli (*Tan et al., 2019*), suggesting that, after SNI, nociceptive input continues to arrive in cortex during NREMS to generate excessive excitation. Indeed, it has been suggested that the SNI model does show spontaneous ectopic electrical activity in peripheral sensory neurons as a result of nerve injury that can increase nociceptive inputs (*Wall and Devor, 1983*; *Devor, 2009*). NREMS is thought to protect relatively weakly from nociceptive inputs (*Claude et al., 2015*), therefore possibly allowing continued processing of spontaneous nociceptive activity that could explain the cortical spectral changes we detected. Besides peripheral nociceptive activity barrages, numerous changes within pain-processing pathways could be involved. We mention that optogenetic stimulation of the thalamic reticular nucleus (TRN) alleviates SNI-related pain (*LeBlanc et al., 2017*). As TRN is involved in the regulation of low-frequency power during NREMS (*Fernandez et al., 2018*), the attenuated delta dynamics observed in NREMS after SNI could result also from altered activity in this nucleus. Pain induces maladaptive plasticity processes with large alteration of brain circuitry, and it is clear further cellular studies will be necessary to understand the origins of elevated arousal frequency in NREMS after SNI before pointing more selectively to one or another mechanism.

The lack of major sleep disruptions after SNI was initially unexpected but seemed in line with other studies. First, we analyzed these animals at a time point when pain from the incision and associated inflammations are largely over (*Guida et al., 2020*), both of which have been shown to disrupt sleep (*Landis et al., 1989*; *Silva et al., 2008*; *Leys et al., 2013*). Moreover, the animals showed a preserved time spent in REMS, suggesting that they did not suffer from chronic mild stress-inflicted sleep disruptions (*Nollet et al., 2019*). However, we did find alterations in a broad range of frequencies including theta power in quiet wakefulness, as reported for other animal models of chronic pain (*LeBlanc et al., 2014*) or for humans with severe neurogenic pain (*Sarnthein et al., 2006*; *Stern et al., 2006*; *Ploner et al., 2017*). Other studies on experimental models of neuropathic pain also report diverse moderate effects on sleep (*Kontinen et al., 2003*; *Leys et al., 2013*). One study on rats 2 and 10 days after SNI suggested that brain states intermediate between NREMS and wakefulness during the resting phase exist (*Cardoso-Cruz et al., 2011*), which could in part reflect our observations. Analysis of sleep disruptions at later stages will help decide whether distinct phases of sleep disruptions mark distinct phases of pain chronicity when anxiety- and depression-related behaviors appear more strongly (*Guida et al., 2020*).

Do animals with SNI suffer from disturbed sleep? Our measures of NREMS's spectral composition point to regionally restricted but tonic imbalances in the contribution of low vs. higher frequencies. Patients with insomnia show such imbalances over widespread brain regions that include sensorimotor areas (*Lecci et al., 2020*). Furthermore, higher power in the beta frequencies has been related to the patients remaining hypervigilant or excessively ruminating at sleep onset (*Perlis et al., 2001*), preventing the deactivation of cortical processes required for the loss of consciousness. Although insomnia also needs subjective assessments that are not possible in animals, this comparison suggests that SNI might suffer from similar experiences due to the tonically enhanced high-frequency oscillations. This interpretation is supported by the elevated wake-up rates in response to mild vibrational stimuli throughout the infraslow cycles, suggesting hyperalertness to environmental disturbance. On top of these tonic changes, there were more frequent cortico-autonomic arousals. Although these do not seem to elevate daytime sleepiness based on the mostly unchanged delta power dynamics across time of day, frequent increases in heart rate during the night could augment cardiovascular risk in the long term (*Silvani, 2019*). To further analyze the animal's conditions during daytime, tests on their cognitive abilities in memory-dependent tasks while locally manipulating sleep in the affected hindlimb

area could be considered. Deficits in working and declarative memories in rodents with SNI have been documented from early periods of chronic pain (*Guida et al., 2020*). Chemogenetic manipulation of neuronal populations proposed to be responsible for the gamma activity in chronic pain, restricted to sleep periods (*Tan et al., 2019*), seems a feasible approach to specifically suppress abnormal pain-related activity during sleep while testing performance in such tasks during wakefulness.

We provide here a guideline to identify arousals in mouse NREMS for which we see the potential to move forward both, animal models of sleep disorders, and advances in the understanding of human sleep disorders. Key for this was the recognition that infraslow electrical variations in the sleeping brain serve as an index for the vulnerability of NREMS to undergo arousal. SNI in mouse qualifies as a rodent model replicating physiological correlates of sleep disorders that are hidden behind a comparatively normal sleep architecture, which is of interest in the field of insomnia (*Lecci et al., 2020*; *van Someren, 2020*). The location of these arousals within the primary cortical sights of pain makes it obvious that the cortical pain axis is causal to sleep disruptions. It could thus be of interest to target these sites specifically to attenuate abnormal activity and to normalize sleep. Addressing the localization and the intensity of local arousals in chronic pain patients could be a first step in the definition of disease subtypes and the severity of their sleep disruption (*Babiloni et al., 2021*). Non-invasive brain stimulation techniques applied locally over sensory or motor cortices have been proposed as therapeutic strategy (*Lefaucheur et al., 2020*). The insights gained from this rodent model of neuropathic pain could thus refine diagnosis and management strategies in the human condition, in which more frequent awakenings are amongst the most consistent effects reported.

# Materials and methods
## Animal housing and experimental groups
Mice from the C57Bl/6J line were singly housed in a temperature- and humidity-controlled environment with a 12 hr/12 hr light-dark cycle (lights on at 9:00 am, corresponding to ZT0, Zeitgeber 0), with access to food and water ad libitum. We first used 36 mice, 10–14 weeks old and bred in our colonies in a conventional-clean animal house, for polysomnography (combined EEG (ECoG)/EMG electrodes), followed by SNI or Sham surgery (18 animals per Sham or SNI group). Mice were transferred from the animal house into the recording room 2–3 days before surgery for polysomnography recording. We recorded a 48 hr-long baseline before SNI or Sham surgeries, followed by recording at 22–23 days after surgery (D20+). These data were used for *Figures 3 and 4*. Total SD in *Figure 4* was done on 24 of these 36 animals (12 SNI, 12 Sham) within 1 day following the recording at D20+. The data for *Figure 1* were obtained from the baseline recordings of 23 randomly selected animals from the previous 36, completed with seven more animals from previous baseline recording in the lab. For EEG (ECoG)/EMG/LFP recordings, 31 C57Bl/6J male mice of the same age were first operated for SNI or Sham (17 and 14, respectively) and 5 days later, implanted for recordings from S1HL (4 Sham, 6 SNI) or PrL (3 Sham, 4 SNI) or both (3 Sham, 4 SNI). The misplaced or non-functional electrodes were excluded. Recordings were carried out from day 20–35 after SNI or Sham surgery. These data were used for *Figures 2, 5 and 6* . The data of 13 animals previously recorded in the lab and otherwise not included in any dataset in this study were used to train the neural network (EEG/EMG implantation, in *Figure 7*). The experiments on sensory-evoked arousals (*Figure 8*) were done on 16 animals (8 Sham, 8 SNI) out of which some (4 sham, 6 SNI) were used for *Figures 5 and 6*. All experimental procedures complied with the Swiss National Institutional Guidelines on Animal Experimentation and the International Association for the Study of Pain (IASP) (*Zimmermann, 1983*). All experiments were approved by the Swiss Cantonal Veterinary Office Committee for Animal Experimentation.

## Surgeries
### Spared nerve injury
The Sham and SNI surgeries were performed as previously described (*Decosterd and Woolf, 2000*; *Bourquin et al., 2006*). Briefly, mice were kept under gas anesthesia (1–2% isofluorane, mixed with $O_2$). The left hindlimb was shaved and the skin incised. The muscles were minimally cut until the sciatic nerve was exposed. Just below the trifurcation between common peroneal, tibial, and sural branches, the common peroneal and tibial branches were tightly ligated together with a silk 6.0 suture and transected. The Sham animals, as controls, went through the same surgery without the transection.

Instead, a 3-mm-long 6.0 silk suture was placed longitudinally to the sciatic nerve. The muscle and the skin were then stitched closed and the animals were monitored via a score sheet established with the Veterinarian Authorities.

## Electrode implantation for polysomnographic and LFP recordings

Surgeries were performed as recently described (*Lecci et al., 2017*; *Fernandez et al., 2018*). Animals were maintained under gas anesthesia (1–2% isoflurane, mixed with $O_2$). Small craniotomies were performed in frontal and parietal areas over the right hemisphere and two gold-plated screws (1.1 mm diameter at their base) (*Mang and Franken, 2012*) were gently inserted to serve as EEG electrodes. Careful scratching of the skull surface with a blade strengthened the attachment of the implant by the glue, so that additional stabilization screws were no longer necessary. Two gold wires were inserted into the neck muscle to serve as EMG electrodes. In the case of LFP recordings, small craniotomies (0.2–0.3 mm) were performed to implant high-impedance tungsten LFP microelectrodes (10–12 MΩ, 75 μm shaft diameter, FHC, Bowdoin, ME) at the following stereotaxic coordinates relative to Bregma in mm, for S1HL: anteroposterior –0.7, lateral –1.8, depth from surface –0.45; for PrL: + 1.8, –0.3, –1.45. For the neutral reference for the LFP recordings, a silver wire (Harvard Apparatus, Holliston, MA) was placed in contact with the bone within a small grove drilled above the cerebellum. The electrodes were then soldered to a female connector and the whole implant was covered with glue and dental cement. The animals were allowed 5 days of recovery, while being monitored via a score sheet established with the Veterinarian Authorities, with access to paracetamol (2 mg/mL, drinking water). The paracetamol was removed when the animals were tethered to the recording cable for another 5 days of habituation prior to the recording.

## Polysomnographic recording

For sleep recordings, recording cables were connected to amplifier boards that were in turn connected to a RHD USB interface board (C3100) using SPI cables (RHD recording system, Intan Technologies, Los Angeles, CA). For EEG/EMG and/or LFP electrodes, signals were recorded through homemade adapters connected to RHD2216 16-channel amplifier chips with bipolar input or RHD2132 32-channel amplifier chips with unipolar inputs and common reference, respectively. Data were acquired at 1000 Hz via a homemade Matlab recording software using the Intan Matlab toolbox. Each recording was then visually scored in 4 s epochs into wake, NREMS, REMS, as described (*Lecci et al., 2017*) using a homemade Matlab scoring software.

## Total sleep deprivation protocol

Total SD was carried out from ZT0 to ZT6 using the gentle handling method used previously in the lab (*Kopp et al., 2006*), while animals remained tethered in their home cage. At ZT3, the cages were changed and, from ZT5 to ZT6, new bedding material was provided. At ZT6, the animals were left undisturbed. The recordings carried out during SD were visually scored to assure the absence of NREMS from ZT0 to ZT6. There were no detected NREMS epochs during SD in the mice included in the analysis.

## Probing sensory arousability with vibration motors

An online detection of continuity and fragility period (described below) was used in a closed-loop manner to time vibration stimuli during NREMS such that sensory arousability could be probed. Small vibrating motors (DC 3–4.2 V Button Type Vibration Motor, diameter 11 mm, thickness 3 mm) were fixed using double-sided tape, at the end of the recording cables, close to the animals' heads. The motors were driven using a Raspberry Pi 3B+ through a 3.3 V pulse-width modulation (PWM) signal. Each motor was calibrated to find the necessary PWM duty-cycle to output the same amount of mild vibration using a homemade vibration measurer equipped with a piezo sensor. A Python script was running on the Raspberry Pi to detect the voltage change sent by the digital-out channels on the Intan RHD USB interface board. Upon detecting a change from low to high, the Python script waited for an additional 4 s and assessed the voltage again. In case the voltage was still high, it launched a 3 s vibration with 25% probability. To close the loop, the PWM signals from the Raspberry Pi driving the motors were as well fed into the analog-in channels of the Intan RHD USB interface board to detect the stimuli time-locked to the EEG/EMG signals. In the experiments, the voltage values were set to

high during either continuity or fragility using online detection as described in Data analysis. Four animals could be tested in parallel for their sensory arousability.

## Histological verification of recording sites

After the in vivo LFP recordings, the animals were deeply anesthetized with pentobarbital (80 mg/kg) and electrode positions were marked through electro-coagulation (50 µA, 8–10 s). The animals were then transcardially perfused with 4% paraformaldehyde (in 0.1 M phosphate buffer). After brain extraction and post-fixation for 24 hr, 100-µm-thick coronal brain sections were cut and imaged in brightfield microscopy to verify correct electrode positioning.

## Data analysis

### Scoring and basic sleep measures

Scoring was done blind to the animal treatment according to standard scoring procedures (*Fernandez et al., 2018*). An MA was scored whenever the EEG presented a desynchronization time matched with a burst of EMG activity lasting maximally four consecutive epochs (16 s). Latency to sleep onset was defined from ZT0 to the first appearance of six consecutive NREMS epochs (24 s). The bout size binning in short, intermediate, and long bouts for NREMS and REMS was obtained from the pulled distribution of the bout sizes from all the animals. The edges of the intermediate bin were defined as: mean – ½ standard deviation to mean + 1 standard deviation.

Spectral power in wakefulness was computed for the raw EEG signal using an FFT on scored 4 s windows after offset correction through subtraction of the mean value of each epoch. The mean power spectrum for QW and AW were normalized by dividing each frequency bin by the sum of the power from 2 to 30 Hz similar to previous work (*LeBlanc et al., 2014*). QW and AW periods were obtained based on the distribution of the natural logarithm of the squared EMG values per 4 s epoch per animal. QW limits were set from 10 to 25 percentile to minimize inclusion of possible periods of drowsiness. Only periods with three or more consecutive epochs (>12 s) were considered. AW limits were set from 50 to 99 percentile to clearly distinguish AW from QW and avoid movement artifacts.

Spectral power for sleep was computed on the raw EEG signal using a FFT on scored 4 s windows after offset correction through subtraction of the mean value of each epoch. The median power spectrum for each state was obtained for epochs non-adjacent to state transitions. The normalization was done through dividing by the average of mean power levels (from 0.75 to 47 Hz) for each vigilance state, ensuring that each state had the same weight in the averaging (*Vassalli and Franken, 2017*). This normalization was done separately for baseline and D20+ recordings. Gamma power at D20+ was extracted through calculating mean power levels between 60 and 80 Hz. Data were normalized to corresponding baseline values.

The heart rate was extracted from the EMG signal as described previously (*Lecci et al., 2017*). Briefly, the EMG signal was highpass-filtered (>25 Hz) and squared. The R peaks of the heartbeats were detected using the Matlab 'Findpeaks' function. Only animals with clearly visible R peaks present in the EMG in NREMS were included in this analysis (*Fernandez et al., 2017*).

For delta power time course, raw delta power (mean power between 1 and 4 Hz from FFT on mean-centered epochs) was extracted for each NREMS epoch non-adjacent to a state transition. Total NREMS time was divided into periods of equal amounts of NREMS (12 in light phases, 6 in dark phases) from which mean values for delta power were computed. The position in time of these periods was not different between groups. Normalization was done via mean values between ZT9 and 12 when sleep pressure is the lowest.

Wake-up and sleep-through events after vibration were scored automatically as follows. For each trial, the EEG and EMG signals were analyzed within time intervals from 5 s prior to 5 s after the vibrations. To distinguish wake-up and sleep-through events, three values were calculated: (1) the ratio theta (5–10 Hz)/delta (1–4 Hz) for the 5 s before stimulation, (2) the difference in the low-/high-frequency ratios (1–4 Hz/100–500 Hz), before and after the stimulation, and (3) the squared EMG amplitude ratio after/before stimulation.

A trial was rejected when the ratio theta/delta was >1 before stimulation or the EMG amplitude was larger before than after stimulation. In this way, trials starting in REMS or wakefulness were excluded. Wake-up events were scored when the difference in low/high ratios mentioned above decreased markedly after stimulation together with EMG activity. Occasionally, some wake-ups were also scored

when EEG or EMG activity was very high while the other channel showed moderate changes. Appropriate thresholds were set upon visual inspection blinded to the animal's condition.

## Analyses related to the 0.02 Hz-fluctuation

### Extraction

The 0.02 Hz-fluctuation in sigma power (10–15 Hz) was extracted from EEG or LFP signals using a wavelet transform (Morlet wavelet, four cycles), calculated over 12 hr recordings in 0.5 Hz bins. The resulting signal was down-sampled to 10 Hz and smoothed using an attenuating FIR filter (cutoff frequency 0.0125 Hz, order of 100, the low order allowing for frequencies above the cutoff). The mean of the datapoints within NREMS and MA epochs was used for normalization (*Figure 1—figure supplement 1D*). The peak and frequency of the 0.02 Hz-fluctuation were calculated through an FFT on continuous NREMS bouts as described (*Lecci et al., 2017*). FFTs from individual bouts at frequency bins from 0 to 0.5 Hz were interpolated to 201 points before averaging across bouts to obtain a single measure per mouse. The angles of the phase of the 0.02 Hz-fluctuation were obtained through the Hilbert transform (Matlab signal processing toolbox). We set the troughs of the 0.02 Hz-fluctuation at 180°, the peaks at 0° (*Figure 1—figure supplement 1G*).

In several instances (*Figure 4D* and *Figure 6*), instead of calculating FFTs in the infraslow frequency range, we needed to detect individual cycles of the 0.02 Hz-fluctuation. To do this, we applied the Matlab 'Findpeaks' function, with the conditions that the peak values were > mean and the trough values < mean, each separated by >20 s. With such parameters, the sequence trough-peak-trough appears only in NREMS and allows to count individual cycles.

### Band-limited power dynamics during the 0.02 Hz-fluctuation

To calculate the power dynamics in different frequency bands, Morlet wavelet transforms were down-sampled to 10 Hz to match the sampling of the 0.02 Hz-fluctuation and normalized by the sum of their means in NREMS. The mean power of each band was then binned in 18 bins of 20°, and a mean across cycles (with or without MAs) of power activity per phase bin was obtained per animal.

### Analysis of activation index

AI was computed by the natural logarithm (ln) of the ratio between beta (16–25 Hz) + low gamma (26–40 Hz) over delta power (1–4 Hz), extracted as described above. Individual cycles from peak to peak were classified whether an MA was present in the fragility period or whether it continued into NREMS. To assess the presence of peaks in activation indices, the 'Findpeaks' function was used at phase values of 90–270°, with mean values used as a threshold.

### Online detection of continuity and fragility periods

For the online detection of fragility and continuity periods during closed-loop sensory stimulation, a homemade software was generated with two layers of decision. The first one determined the likely current state of vigilance (wake, NREMS, or REMS), whereas the second one made a machine learning-based decision between a continuity or a fragility period.

(1) Determination of vigilance state. This assessment was based on power band ratios characteristic for wake, NREMS, and REMS using appropriate thresholds (*Figure 7B and C*). Every second, an FFT was calculated on the mean-centered last 4 s of EEG values and the power ratio between the delta (1–4 Hz) and the theta (5–10 Hz) was calculated.

#### Transitions out of wake

(1) Switch to NREMS if the last 3 s of EMG were below a high threshold and at least two out of the three last seconds of ratio were above a high threshold. (2) Switch to REMS if the full 5 s of EMG were below a low threshold and the full 5 s of ratio were above a high threshold.

#### Transitions out of NREMS

(1) Switch to wake if the last second of EMG was above a high threshold. (2) Switch to REMS if the last 5 s of EMG were below a low threshold and if among the last 5 s of ratio, at least four were below a low threshold and all five were below a high threshold.

#### Transitions out of REMS

(1) Switch to wake if the last second of EMG were above a high threshold. (2) Switch to NREMS if the ratio was above a high threshold for at least 4 out of 5 s.

(2) Continuity and fragility detection. From the previous step, the value of sigma power (10–15 Hz) was kept every second. The mean sigma value in NREMS was dynamically updated if the likely state was determined as NREMS and used to normalize the incoming sigma power values. The last 200 s of sigma power regardless of the likely state were kept in memory. We heuristically found that a ninth-order polynomial fit (Matlab 'polyfit' and 'polyval' functions) best approximated the 0.02 Hz-fluctuation. To train the network, we first generated online-estimated 0.02 Hz-fluctuation at 1 Hz for the 12 hr of the light phase. We next applied offline cycle detection in NREMS periods. For simplicity, and in agreement with previous measures of sensory arousability (*Lecci et al., 2017*), we set the continuity periods from trough to peak and fragility from peak to trough. Then, we subdivided these recordings in chunks of 200 s (moving window of 1 s, as they would appear online) while assigning the label continuity, fragility, or none. We could thus obtain 43,000 labeled chunks per 12 hr of recording. We used 642,000 of these chunks from 13 animals to train a neural network (pattern recognition 'nprtool' from Matlab Statistics and Machine Learning Toolbox) 70% for training, 15% for validation, and 15% for testing. The network was composed of 1 hidden layer with 10 neurons and 1 output layer with the 3 different outputs. We then used the generated neural network online to take the decision between continuity, fragility, or none.

### Statistics

The statistics were done using Matlab R2018a and the R statistical language version 3.6.1. The normality and homogeneity of the variances (homoscedasticity) were assessed using the Shapiro–Wilk and the Bartlett tests, respectively, to decide for parametric statistics. In the cases where normality or homoscedasticity were violated, a log transformation was considered first and, finally, non-parametric post-hoc tests were used (Wilcoxon rank-sum test for unpaired and signed-rank test for paired data). The degrees of freedom and residuals for the F values are reported according to the R output. Post-hoc analyses were done only when the interaction between factors was significant ($p<0.05$). Bonferroni's correction for multiple comparisons was applied routinely, and the corrected α values are given in the legends. All statistical tests, factors used in the ANOVAs, and corresponding Cohen's effect size are given in *Supplementary file 1* for every figure panel. The factors used in the ANOVAs are mentioned in the figures once the corresponding effects were significant. The interaction is denoted as X. The factors used in the analysis were: 'treatment' with two levels: Sham and SNI; 'day' with two repeated levels: baseline and D20+; 'size' with three repeated levels: small, intermediate, or long bouts; 'period' with two repeated levels: continuity or fragility; 'SD' with two repeated levels: control or recovery after sleep deprivation; 'state' with three repeated levels: wake, NREMS, or REMS; 'MAs' with two levels: with or without MA in the fragility period; 'peak' with two repeated levels: cycles with or without a peak in AI during fragility periods. The circular statistics were done using the CircStat for Matlab toolbox (*Berens, 2009*).

### Acknowledgements

All lab members provided critical input at all stages of this manuscript. The excellent animal caretaking headed by Michelle Blom and the Team of Animaliers, in particular Titouan Tromme, is highly appreciated. Expert veterinary support and advice was provided by Drs. Gisèle Ferrand and Laure Sériot. We thank Christiane Devenoges for support in histological analysis and Marie Pertin and Guylène Kirschmann for surgical support with SNI. Dr. Simone Astori and Dr. Marc Suter provided insightful comments on pre-final versions of the manuscript and Laura Solanelles Farré helped with careful proofreading. The useful discussions with Raquel Sandoval Adaia, Paul Franken, Thomas Nevian, Francesca Siclari, and Raphaelle Winsky-Sommerer are gratefully acknowledged. This study was funded by The Swiss National Science Foundation (no. 310030_184759 to AL, no. 310030_179169 to ID, no. 320030_179194 to SF), and Etat de Vaud.

## Additional information

### Funding

| Funder | Grant reference number | Author |
|---|---|---|
| Swiss National Science Foundation | 310030_184759 | Anita Lüthi |
| Swiss National Science Foundation | 310030_179169 | Isabelle Decosterd |
| Swiss National Science Foundation | 320030-179194 | Stephany Fulda |
| Etat de Vaud | | Anita Lüthi |

The funders had no role in study design, data collection and interpretation, or the decision to submit the work for publication.

### Author contributions

Romain Cardis, Conceptualization, Data curation, Formal analysis, Investigation, Methodology, Software, Visualization, Writing – review and editing; Sandro Lecci, Conceptualization, Data curation, Formal analysis, Methodology; Laura MJ Fernandez, Data curation, Methodology; Alejandro Osorio-Forero, Methodology; Paul Chu Sin Chung, Formal analysis, Writing – review and editing; Stephany Fulda, Conceptualization, Validation, Writing – review and editing; Isabelle Decosterd, Conceptualization, Data curation, Funding acquisition, Project administration, Validation, Visualization, Writing – original draft, Writing – review and editing; Anita Lüthi, Conceptualization, Data curation, Funding acquisition, Project administration, Supervision, Validation, Visualization, Writing – original draft, Writing – review and editing

### Author ORCIDs

Romain Cardis ⓘ http://orcid.org/0000-0002-0595-4175
Laura MJ Fernandez ⓘ http://orcid.org/0000-0002-7942-3369
Alejandro Osorio-Forero ⓘ http://orcid.org/0000-0003-4341-4206
Stephany Fulda ⓘ http://orcid.org/0000-0001-8416-2610
Isabelle Decosterd ⓘ http://orcid.org/0000-0002-4820-5289
Anita Lüthi ⓘ http://orcid.org/0000-0002-4954-4143

### Ethics

All experimental procedures complied with the Swiss National Institutional Guidelines on Animal Experimentation and were approved by the Swiss Cantonal Veterinary Office Committee for Animal Experimentation.

### Decision letter and Author response

Decision letter https://doi.org/10.7554/eLife.65835.sa1
Author response https://doi.org/10.7554/eLife.65835.sa2

## Additional files

### Supplementary files

• Supplementary file 1. Table for complete statistical information for all figure panels, arranged according to figure panel, statistical test used, group size, value of test statistics, p values, and Cohen's effect sizes. We have instead removed this info from all figure legends to simplify reading.

• Transparent reporting form

• Source code 1. Extraction of individual cycles of the 0.02 Hz-fluctuation.

• Source code 2. Analysis of transitions out of fragility periods, characterized by Hilbert angles from 90 to 270°.

• Source code 3. Hilbert phase analysis of the 0.02 Hz-fluctuation.

• Source code 4. Extraction of 0.02 Hz-fluctuation from one non-rapid eye movement sleep bout at 10 Hz resolution.

• Source code 5. Wavelet analysis used for calculation of sigma power dynamics, used in Source code 4.

## Data availability

All processed data generated or analyzed during this study are included in the manuscript and supporting files. Source data files are provided for all figures. Matlab codes for major analyses are provided.

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
