## [Decision Letter]

**Acceptance summary:**

The manuscript addresses the relationship between chronic pain and the patterns and quality of sleep in mice. The study employs high quality EEG analyses to show that although mice with neuropathic pain retain general sleep architecture, they are more easily aroused in sleep and show alterations in infraslow oscillatory activity in the brain in conjunction with behavioral abnormalities. The findings provide insights into heightened arousability during sleep in pain and bad sleep quality as a comorbidity of chronic pain.

**Decision letter after peer review:**

Thank you for submitting your article "Cortico-autonomic local arousals and heightened somatosensory arousability during NREM sleep of mice in neuropathic pain" for consideration by *eLife*. Your article has been reviewed by 2 peer reviewers, and the evaluation has been overseen by a Reviewing Editor and Christian Büchel as the Senior Editor. The following individual involved in review of your submission has agreed to reveal their identity: Carl Y Saab (Reviewer #2).

Essential revisions:

1) Although the study reports interesting data on alterations in sleep in chronic pain, the functional relationship between pain and sleep is not addressed. In order to enable a significant scientific advance, behavioural data on pain in conjunction with altered sleep are necessary.

2) There are a number of concerns about the analysis of the EEG data. A significant reanalysis of EEG data is necessary and it is recommended to address all points related to analysis and interpretation of data on oscillatory activity.

*Reviewer #1:*

In the present manuscript, the authors study the relationship of chronic pain and sleep in mice. Specifically, they first established a mouse model of neuropathic pain and show that the general sleep architecture is fully preserved in mice. Then using mechanicovibrational stimulation, they show that these mice are easier arousable in sleep. Critically, they link this phenomenon to an infraslow (approx. 0.02 Hz, every 50s) rhythm. The findings are being interpreted as heightened arousability in pain during sleep, which might constitute a promising target for future interventions.

The paper is very well written, has great illustration and nicely guides the reader from the initial hypothesis to the main findings. It provides an interesting link between sleep physiology and pain perception. While overall very convincingly written, several conceptual and methodological questions remain:

1. Why is the increased arousability only linked to mechanical stimulation? Wouldn't one expect that neuropathic pain patients suffer more constantly and are thus, more likely to experience sub-optimal night sleep with more microarousals?

2. How can the authors claim specificity for the infraslow oscillation when no alternative frequency bands or spectral phenomena were probed? The general link to the 0.02 Hz oscillations is interesting, but for the naïve reader the step from Figure 1 to Figure 2 is hard to understand: Besides that they authors first described the 0.02 Hz oscillation in the Lecci 2017 paper, there is no direct obvious link to Figure 1.

In particular, since Figure 1 indicates no differences between groups. Hence, any other spectral metric could have been investigated. Maybe the authors could elaborate on the reasoning here.

3. How do the authors interpret the fact that the observed effects (e.g. Figure 3) are not frequency specific, but seem to be present in almost every frequency band? Overall, the presentation of the results would benefit from clearly separating physiology vs. pathophysiology-based assumptions. Right now, the manuscript is difficult to follow because both a presented in an interleaved manner and one wonders which effects are truly specific to the pain model.

4. The visualization of the coupling in Figure 5 appears very cluttered. Could the authors maybe include circular representations of the data in addition to the histogram-based ones (with phase on the x-axis)? Specifically to avoid the binary contrasts between two large (270-90{degree sign} and 90-270{degree sign}) bins that were used for statistical comparisons? Why not utilize appropriate statistics that do not require rendering the data into linear format, such as Watson-Williams or Harrison-Kanji tests when testing for coupling interactions.

This seems critical, since the authors essentially present a cross-frequency coupling analysis (0.02 Hz to power or arousal coupling), which might benefit from the guidelines formulatedby Aru et al. (2015) Current Opinion in Neurobiology

5. Interpretation of oscillatory signatures is sometimes a bit unspecific. E.g. in the introduction they authors discuss 'enriched oscillations', i.e. indexing increased spectral power – but how would they disentangle that from e.g. movement artifacts, which cover multiple frequency bands after spectral transformations, which is the defining element of microarousals?

6. Overall, the manuscript is difficult to read, because lots of different analyses are being introduced and detailed, but it is difficult to extract the main gist why the specific analysis was carried out.

7. Queries per specific Figures:

Figure 1A – Why is more time spent awake in the 'dark phase'? Shouldn't this be the other way round?

Figure 1E – How were the spectra normalized?

Figure 4F-H implies a non-frequency specific effect, why not quantify and display the true comodulation to assess the spectral extent?

Figure 6 – Isn't the presented analysis circular in nature? The authors predict labels using machine learning based on already classified stages using mutually-dependent criteria.

Figure 7 – A Visualization of when stimulation is most effective in controls would help.

8. Please report t values and effect sizes and not just p-values, specifically in the first part of the manuscript.

9. When discussing their finding 1 in the main discussing: What is the early stage of the disease and to what data are the authors referring to?

Taken together, the authors present an interesting study with multitude of results using a variety of analyses. Overall, this manuscript provides a better understanding for sleep physiology by further adding evidence for the importance of the 0.02 Hz oscillation. In addition, it provides an interesting perspective on pain and sleep. However, in its present form it is difficult to extract the main findings.

Overall, this is an interesting paper well within the scope of *eLife*, which requires several clarifications, but is a strong candidate for publication.

10. The authors should try to clearly separate hypotheses, predictions and analyses concerning physiology vs. pathophysiology. In its present form both appear interleaved, making it very difficult to follow the manuscript and reasoning.

*Reviewer #2:*

Rigorous investigation of potential correlation between chronic pain and sleep disorder in a mouse model of spared sciatic nerve injury (SNI). Methodology includes conventional sleep analysis using EEG, EMG and a novel arousal assay within epochs referred to as 0.02 Hz fragility-continuity during light phase NREM. No difference was reported in global sleep architecture. However, mice with SNI manifest increased cortical fragility in somatosensory hindlimb (S1) contralateral to SNI, but not in ipsilateral S1 or prelimbic cortex, compared to sham. These heart changes also correlated with increased rate. Moreover, mice with SNI were hypersensitive to spontaneous arousal, as well as arousal evoked by mechano-vibration timed to fragility periods in a closed-loop manner.

Strengths:

The topic of chronic pain, brain abnormality, EEG and sleep is timely.

Very well written manuscript, logical flow, clear diagrams.

Methodology is robust, uses conventional techniques and develops a new assay to investigate fragility-continuity that enabled novel findings. This novel assay is potentially useful to sleep researchers.

Conclusions are conservative and warranted by the data and the statistics.

Literature is comprehensively reviewed.

Weaknesses:

1. Study reads as if it is designed to explore sleep differences in SNI mice, rather than to test a hypothesis based on a clearly defined neurophysiological mechanism. There is no mechanistic evidence (other than a speculation about sustained peripheral drive from injury site, which is more stochastic than regular).

Significant findings have considerable limitations. They apply only to NREMS, during light phase, in absence of EMG, and in S1. This raises concerns about relevance and validity, especially that one animal model was used, no evidence for pain presented (and if so what pain category or correlation with pain severity), one time point at D20+, changes are minor, and other cited studies have noted local cortical arousal in humans (Nobili 2011).

2. Link between pain and sleep is bi-directional, as stated by the authors, but pain is causal to sleep disorder, so recommendations to better understand sleep to treat pain should be reworded more carefully. Also recommendations for treatment options (e.g. TMS localized to S1) should be re-considered, especially when mechanisms are still unknown.

3. Analysis of EEG data requires more clarification. Normalization of EEG is common practice, though standards differ, along with data interpretation. Due to the inherent variability in EEG, the most accurate analysis is within-subject comparison. In this study, EEG was normalized dividing by the average mean power from 0.75-47 Hz. This could lead to bias. For e.g., if absolute EEG power in the 1-45 Hz is modulated in one group, normalization will dilute this modulation and artificially enhance power in bands outside this range.

4. Movement artifact is a common source of noise in EEG. How was this dealt with? It is recommended that an automated algorithm be used to limit subjectivity and bias.

5. Authors conclude no change in theta power in awake mice with SNI. However, cited paper by Leblanc 2014 clearly showed increased theta in awake rats after CCI (another sciatic pain model). The same group has since published 7 papers replicating these findings in multiple pain models, including one study in humans. Thalamic mechanisms for theta was presented by Leblanc, as correctly discussed by the authors (however that was Leblanc 2017, not 2014, please correct). The only explanation for the discrepancy here is the fact that Leblanc recorded theta during brief (~15 min) awake resting state with no apparent movement once per day. Analysis of EEG during the entire wake period when an animal is engaged in day-time activity (potentially distracted from pain) might dilute this effect on theta power.

6. Following the same logic, it should also be noted that the increased gamma reported by Tan 2019 refers to the CFA model of acute inflammatory pain, which is different from chronic neuropathic pain.

7. The study is largely based on the premise that mice are in chronic pain at D20+ after SNI. Typically, studies using animal models include behavioral data to verify the pain state (for e.g. sensory hypersensitivity, etc.) Whereas SNI is a widely used model, verification of model reproducibility is essential. Moreover, it appears from the literature that the predominant evidence for pain in this model is sensory hypersensitivity evoked by mechanical and thermal stimuli. Do we know if rodents (particularly mice) with SNI manifest spontaneous pain? This is important because only one model is used in this study and at only one time-point. This is also relevant when making translational claims about human pain, which is multi-dimensional (reference to model mimicking human pain should be avoided).

8. Authors cite several studies showing abnormal findings in alpha, beta and low-gamma during sleep (p4). These findings seem inconsistent with this paper. Please discuss (also make sure you're making apples-to-apples comparisons, including epoch lengths, normalization, model, etc.)

9. Figure legends too crowded with statistics.

10. Some icons in legends hard to interpret (e.g. Figure 1G).

11. What are units in Figure 1F?

---

## [Author Response]

Essential revisions:1) Although the study reports interesting data on alterations in sleep in chronic pain, the functional relationship between pain and sleep is not addressed. In order to enable a significant scientific advance, behavioural data on pain in conjunction with altered sleep are necessary.

This is an important remark that is clearly essential for this study. We provide here two novel datasets for this outstanding issue that helps resolve it in two complementary aspects.

i. We present behavioral responses to mechanical and thermal (heat) stimulation after SNI and sham surgeries in mice. The data are from another ongoing experiment in our laboratory, and they include measurements at times points similar to the present study. In our hands, and in the hands of many other labs, all nerve-injured animals with the well performed SNI technique develop tactile and thermal allodynia/hyperalgesia-like behavior (for review, see Guida *et al.*, 2020; doi:10.3390/ijms21093396).These data are included in the rebuttal to the concerns of Reviewer 2.

ii. We re-analyze the data from the mice included in this study regarding the presence of elevated theta and high-frequency power. Such elevations have been repeatedly linked to pain hypersensitivity in experimental models and described as an “electroencephalographic signatures of pain” by LeBlanc et al. (2017). They are found in acute and chronic experimental models and respond to analgesics. When suppressed by optogenetic manipulation of thalamic activity in mice, or via thalamic lesions in humans, pain relief has been observed (Sarnthein et al., 2006) (LeBlanc et al., 2016) (LeBlanc et al., 2017). We do find changes in the power spectrum of the SNI groups during quiet wakefulness (a state determined polysomnographically), which strongly supports the presence of neuropathic pain-like hypersensitivity in the animals used in our study. These data are included in a new Figure 2 in the main manuscript.

In conclusion, we now provide the most sensitive physiological measures of experimental chronic pain currently available. In particular, we were able to correlate it with the sleep data, as requested by the Editor.

2) There are a number of concerns about the analysis of the EEG data. A significant reanalysis of EEG data is necessary and it is recommended to address all points related to analysis and interpretation of data on oscillatory activity.

In response to this essential revision request, we have carried out all the analyses requested by the reviewers. We provide detailed answers to these in a point-to-point manner.

Reviewer #1:[…] The paper is very well written, has great illustration and nicely guides the reader from the initial hypothesis to the main findings. It provides an interesting link between sleep physiology and pain perception. While overall very convincingly written, several conceptual and methodological questions remain:1. Why is the increased arousability only linked to mechanical stimulation? Wouldn't one expect that neuropathic pain patients suffer more constantly and are thus, more likely to experience sub-optimal night sleep with more microarousals?

The increased arousability is linked to both – mechanical stimulation on the skull, and a higher frequency of brief arousals that we characterize as “cortico-autonomic arousals” in the title. These are not the usual microarousals commonly described in mice. Instead, these are cortical arousals occurring locally in the cortex but that are nevertheless accompanied by heart rate increases. They take place in the hindlimb primary somatosensory cortex that is linked to the discriminative sensory component of pain processing (localization, intensity and modality).

We conclude that nerve injury, and thus neuropathic pain-like alterations, augments both, arousability to external mechanical stimulation and spontaneous arousability in precisely those brain areas that are concerned with pain processing.

2. How can the authors claim specificity for the infraslow oscillation when no alternative frequency bands or spectral phenomena were probed? The general link to the 0.02 Hz oscillations is interesting, but for the naïve reader the step from Figure 1 to Figure 2 is hard to understand: Besides that they authors first described the 0.02 Hz oscillation in the Lecci 2017 paper, there is no direct obvious link to Figure 1.In particular, since Figure 1 indicates no differences between groups. Hence, any other spectral metric could have been investigated. Maybe the authors could elaborate on the reasoning here.

We are grateful for this comment that made us realize that we have to describe more in details the reasoning for the use of the infraslow oscillation. We do this now explicitly in the rewritten Introduction (3^rd^ paragraph), in which we review evidences that the infraslow time scale stands out in terms of identifying moments of sleep vulnerability.

Regarding the link from Figure 1 to Figure 2 that the reviewer describes as hard to understand: The figure sequence has now been changed such that the physiology of the infraslow oscillation comes first, and is then followed by studies in the experimental model of neuropathic pain (SNI).

Together, with these re-arrangements, we believe we have solved the major concerns of the reviewer: the issue of specificity, the clarity of the figure arrangement, and the reasoning for the choice of the infraslow oscillation.

3. How do the authors interpret the fact that the observed effects (e.g. Figure 3) are not frequency specific, but seem to be present in almost every frequency band? Overall, the presentation of the results would benefit from clearly separating physiology vs. pathophysiology-based assumptions. Right now, the manuscript is difficult to follow because both a presented in an interleaved manner and one wonders which effects are truly specific to the pain model.

We are not sure which observed effects in Figure 3 the reviewer refers to. The data in this figure do not concern frequency specificity. In case the reviewer thought about Figure 4, then we remark the following: the observed effects cover precisely the frequency bands that need to change to give rise to an arousal: low-frequency bands decrease in power, while high-frequency bands increase.

We have taken up the reviewer’s excellent suggestion to better separate physiology versus pathophysiology. To do so, we position the Figure describing the phase-coupling of microarousals in healthy mice to the infraslow oscillation as the new Figure 1 (original Figure 2). We additionally modified the figure to make the infraslow oscillation more clear. Then, we introduce a new Figure 2 describing the physiological correlates of pain signature in SNI operated mice. The remaining figures are then focused on sleep pathophysiology. In this way, the separation between physiology and pathophysiology should be clear.

4. The visualization of the coupling in Figure 5 appears very cluttered. Could the authors maybe include circular representations of the data in addition to the histogram-based ones (with phase on the x-axis)? Specifically to avoid the binary contrasts between two large (270-90{degree sign} and 90-270{degree sign}) bins that were used for statistical comparisons? Why not utilize appropriate statistics that do not require rendering the data into linear format, such as Watson-Williams or Harrison-Kanji tests when testing for coupling interactions.This seems critical, since the authors essentially present a cross-frequency coupling analysis (0.02 Hz to power or arousal coupling), which might benefit from the guidelines formulatedby Aru et al. (2015) Current Opinion in Neurobiology.

We have simplified the presentation of Figure 5 (now the new Figure 6) to amend the reviewers’ concern. The changes involve:

– The data of the original panels 5 H and I are now moved to the supplemental figure 2, panels A-C;

– The data from the original panel J are now shown more expanded in the new panel 6 H;

– The data from the original panels K, L are now shown expanded in the new panels 6 I, J.

Regarding the use of circular representations, we agree that this could be a worthwhile alternate way of presenting data that involve phase relationships. We are actually using this kind of representation in the lab (as recently done for the tuning of head direction units in the 360 ° space, see Vantomme et al., 2020, doi: 10.1016/j.celrep.2020.107747). Regarding the infraslow oscillation, we would prefer to keep a linear presentation that we have already started using some years ago (Lecci et al., 2017, doi: 10.1126/sciadv.1602026) when we first reported on the infraslow oscillation. Furthermore, we find the representation of the microarousal clustering around the trough of the oscillation (see new Figure 1) clearer as it brings the importance of the trough of the oscillation for sleep’s vulnerability to the forefront.

This representation also makes it clear that we are indeed interested in the two large bins described by the reviewer. These bins are defined by the natural clustering of microarousals. We currently have no incentive to use a finer graduation of angles as the microarousal distribution does not give us a physiological reason to do so.

5. Interpretation of oscillatory signatures is sometimes a bit unspecific. E.g. in the introduction they authors discuss 'enriched oscillations', i.e. indexing increased spectral power – but how would they disentangle that from e.g. movement artifacts, which cover multiple frequency bands after spectral transformations, which is the defining element of microarousals?

The term “enriched oscillations” was used in the context of the description of insomnia patients. This part has been removed and the term “enriched oscillations” is no longer present.

Regarding movement artifacts.

The Intan recording system we used to obtain the data presented in this study renders movement artefacts very small. The reason is that the signals are digitized at the level of the pre-amplifier that is present on the animals head. In this way, movements that primarily arise from the movement of the cables are eliminated. Please also realized that when we study the novel cortico-autonomic arousals the EMG is completely flat (panel C of the new Figure 6). Therefore, in the analysis of these novel events, movement artefacts are not a potential source of confound. Similarly, when we analyze the spectral composition of quiet wakefulness, the EMG levels are very low (see the new Figure 2, panel C).

More generally, movement artefacts become a minor concern once recordings are obtained intracranially, as is the case for our LFP recordings. The problem is much greater in the case of surface EEG recordings, as is frequently done in humans. In our experiments, EEG signals were only used for scoring of microarousals and for the extraction of the infraslow dynamics in Figure 1. For both these analyses, movement artefacts played no role.

6. Overall, the manuscript is difficult to read, because lots of different analyses are being introduced and detailed, but it is difficult to extract the main gist why the specific analysis was carried out.

We appreciate this comment because it helped us realize the need of improving the clarity and structure of the manuscript. As already described, the work is now separated into sleep physiology- and abnormal sleep related to an experimental model of neuropathic pain. The transition between these parts is more explicitly described and so are the analyses.

7. Queries per specific Figures:Figure 1A – Why is more time spent awake in the 'dark phase'? Shouldn't this be the other way round?

Mice are nocturnal animals. Therefore, they spent most of their time awake during the dark phase (night), and most of their time asleep during the light phase (day).

Figure 1E – How were the spectra normalized?

We clearly describe the normalization procedure in the Methods on p.30-31 of the clean manuscript (lines 638-645). For the spectra of quiet wakefulness, we followed the procedure specified in the papers of C. Saab and colleagues, 2014, doi:10.1016/j.pain.2014.01.013 to enable direct comparisons between models (p.30, lanes 630-637).

Figure 4F-H implies a non-frequency specific effect, why not quantify and display the true comodulation to assess the spectral extent?

We have already explained above that we motivate our choice of analysis based on the natural physiology of how microarousals clusters around the trough of the infraslow oscillation. Our major questions were: what are the spectral changes at these moments of high sleep vulnerability? Do they coincide with power changes in frequency patterns that coincide with an arousal? Our results show that this is indeed the case: at moments of elevated sleep vulnerability, LFP data collected in the S1 hindlimb cortex show a pattern of frequency changes that indicate that an arousal takes place. An arousal is not a frequency-specific change, it consists of a decrease in low- and an increase in the power of higher frequencies. This is exactly what we see.

Figure 6 – Isn't the presented analysis circular in nature? The authors predict labels using machine learning based on already classified stages using mutually-dependent criteria.

For this analysis, we first train the algorithm on datasets that are already scored. It is only then that the online monitoring of continuity and fragility periods is done. Thus, the presented analysis is not circular. The training was only done to enable the machine to predict the continuity and fragility periods online. No information about the MAs themselves was given during the training of the network. Moreover, the training was done on completely separate animals compared to the one used in the paper.

Figure 7 – A Visualization of when stimulation is most effective in controls would help.

We are not sure what the reviewer means by most effective. In this figure (new Figure 8), we clearly indicate two representative traces of wake-up and sleep-through (panel C). Moreover, using color-coded arrows, we present several NREMS bouts and indicate when stimulation resulted in a wake-up or a sleep-through.

8. Please report t values and effect sizes and not just p-values, specifically in the first part of the manuscript.

Please find now a new Supplementary Table for Statistics, arranged according to figure panel, statistical test used, group size, value of test statistics, p values and Cohen’s effect sizes. We have instead removed this info from all figure legends to simplify reading.

9. When discussing their finding 1 in the main discussing: What is the early stage of the disease and to what data are the authors referring to?

We study sleep 22-23 days after SNI surgery. In this model, pain hypersensitivity is initiated during the first 3 days after surgery, reaches a plateau a week after and is stable for weeks, even months at. Anxiety- and depressive-like behavior as well as cognitive impairment are seen during late phases. Three weeks after SNI, peripheral abnormal activity and central sensitization mechanisms responsible for pain hypersensitivity are established. On the other hand, at this time point, there is only low level of psychopathology associated with peripheral nerve injury and cognitive bias (see review of Guida et al., 2020 doi:10.3390/ijms21093396). In addition, the absence of an increased REMS in our animals (seen in Nollet et al., 2019, doi:10.1073/pnas.1816456116) also tends toward the conclusion that nerve-injury related pain is predominant during this time frame.

10. The authors should try to clearly separate hypotheses, predictions and analyses concerning physiology vs. pathophysiology. In its present form both appear interleaved, making it very difficult to follow the manuscript and reasoning.

With the above-mentioned points, we hope to have responded to these recommendations so that the paper is now easy to follow in terms of reading and reasoning.

Reviewer #2:[…] 1. Study reads as if it is designed to explore sleep differences in SNI mice, rather than to test a hypothesis based on a clearly defined neurophysiological mechanism. There is no mechanistic evidence (other than a speculation about sustained peripheral drive from injury site, which is more stochastic than regular).Significant findings have considerable limitations. They apply only to NREMS, during light phase, in absence of EMG, and in S1. This raises concerns about relevance and validity, especially that one animal model was used, no evidence for pain presented (and if so what pain category or correlation with pain severity), one time point at D20+, changes are minor, and other cited studies have noted local cortical arousal in humans (Nobili 2011).

We thank the reviewer for this detailed summary that clearly lays out the revisions required to improve the strengths of the paper. We hope that with the revisions added, we have amended the major concerns. Please note in particular that we now do present evidence that mice might be suffering from pain electroencephalography evidences. We have now also expanded the analysis beyond NREMS to active and quiet wakefulness, alleviating the concern of the limited significance.

2. Link between pain and sleep is bi-directional, as stated by the authors, but pain is causal to sleep disorder, so recommendations to better understand sleep to treat pain should be reworded more carefully. Also recommendations for treatment options (e.g. TMS localized to S1) should be re-considered, especially when mechanisms are still unknown.

We agree that we have kept the text on the implications of our findings for treatment options short and over-conclusive on the dimension of sleep intervention as a new approach for pain alleviation. We have now formulated a novel paragraph at the end of the discussion to discuss this topic – we discuss both, the implications for animal models of sleep disorders, and for diagnosis and management strategies in patients. In addition, bad sleep decreases highly the quality of life in patients suffering from neuropathic pain. A large subset of patients have an intractable form of neuropathic pain, and improving quality of life by reducing pain interferences such as sleep and mood disorders is essential for their treatment. In this line, we think that we point out early mechanisms for sleep disturbances that can be further explored to understand better their contribution to the persistence of pain and development of other related alterations.

3. Analysis of EEG data requires more clarification. Normalization of EEG is common practice, though standards differ, along with data interpretation. Due to the inherent variability in EEG, the most accurate analysis is within-subject comparison. In this study, EEG was normalized dividing by the average mean power from 0.75-47 Hz. This could lead to bias. For e.g., if absolute EEG power in the 1-45 Hz is modulated in one group, normalization will dilute this modulation and artificially enhance power in bands outside this range.

We appreciate the reviewers’ concerns regarding normalization. Normalization of EEG power spectra is common practice. Normalization removes the variations in absolute power values between individuals that naturally arise from the different positioning and/or impedance of recording electrodes with respect to the brain and from variations in brain size. Moreover, normalization permits to see directly how much power changes in one frequency band relative to another one. This is exactly the information that is needed to estimate chronic alterations in brain states after in SNI in mice.

What is critical is how normalization is done. In our case, we used the 0.75-47 Hz band to determine relative power changes between vigilance states. When it came to the quantification of gamma power that lies outside this band, we used only power value ratios between baseline and after SNI within individuals. Therefore, there were no artificial enhancements of power values.

For the quantification of changes in theta power in quiet wakefulness, we used normalization procedures described previously for the chronic constrictive model of neuropathic pain (CCI) model and other pain models (LeBlanc et al., 2014, doi:10.1016/j.pain.2014.01.013) in the interest of comparison (see also response to reviewer 1 and below). This consisted of normalizing the spectra by their mean power between 2-30 Hz.

4. Movement artifact is a common source of noise in EEG. How was this dealt with? It is recommended that an automated algorithm be used to limit subjectivity and bias.

We have received a similar comment from Reviewer 1. Therefore, we reply here in the same words.

The Intan recording system we used to obtain the data presented in this study renders movement artefacts very small. The reason is that the signals are digitized at the level of the pre-amplifier that is present on the animals head. In this way, movements that primarily arise from the movement of the cables are eliminated. Please also realized that when we study the novel cortico-autonomic arousals the EMG is completely flat (panel C of the new Figure 6). Therefore, in the analysis of these novel events, movement artefacts are not a potential source of confound. Similarly, when we analyze the spectral composition of quiet wakefulness, the EMG levels are very low (see the new Figure 2, panel C).

More generally, movement artefacts become a minor concern once recordings are obtained intracranially, as is the case for our LFP recordings. The problem is much greater in the case of surface EEG recordings, as is frequently done in humans. In our experiments, EEG signals were only used for scoring of microarousals and for the extraction of the infraslow dynamics in Figure 1. For both these analyses, movement artefacts played no role.

5. Authors conclude no change in theta power in awake mice with SNI. However, cited paper by Leblanc 2014 clearly showed increased theta in awake rats after CCI (another sciatic pain model). The same group has since published 7 papers replicating these findings in multiple pain models, including one study in humans. Thalamic mechanisms for theta was presented by Leblanc, as correctly discussed by the authors (however that was Leblanc 2017, not 2014, please correct). The only explanation for the discrepancy here is the fact that Leblanc recorded theta during brief (~15 min) awake resting state with no apparent movement once per day. Analysis of EEG during the entire wake period when an animal is engaged in day-time activity (potentially distracted from pain) might dilute this effect on theta power.

This is an excellent point raised by the reviewer that we have now followed up using an analysis of quiet wakefulness. The results are now included (see figure summarized in a new Figure 2. Briefly, after SNI, mice do show an enhanced power at frequencies above ~5 Hz). When binned in frequency bands for theta (5-10 Hz), σ (10-15 Hz), β (16-25 Hz) and low (26-40 Hz) and high (60-80 Hz), we find significant increases in theta, beta and low gamma power.

We now cite adequately the LeBlanc studies.

6. Following the same logic, it should also be noted that the increased γ reported by Tan 2019 refers to the CFA model of acute inflammatory pain, which is different from chronic neuropathic pain.

We now appropriately cite this reference.

7. The study is largely based on the premise that mice are in chronic pain at D20+ after SNI. Typically, studies using animal models include behavioral data to verify the pain state (for e.g. sensory hypersensitivity, etc.) Whereas SNI is a widely used model, verification of model reproducibility is essential. Moreover, it appears from the literature that the predominant evidence for pain in this model is sensory hypersensitivity evoked by mechanical and thermal stimuli. Do we know if rodents (particularly mice) with SNI manifest spontaneous pain? This is important because only one model is used in this study and at only one time-point. This is also relevant when making translational claims about human pain, which is multi-dimensional (reference to model mimicking human pain should be avoided).

This is an important question and we divide our answer in several parts.

Regarding the need of behavioral data: We have intentionally refrained from testing the animals for recording stimulus-evoked pain hypersensitivity. First, we have large evidence in our lab that all operated mice develop pain-related behavior. Second, we know that behavioral testing of pain sensitivity is stressful for the animals (the tests induce nocifensive behavior only in SNI animals plus testing itself is disrupting the animal). It can likely be a cause of augmented sleep deprivation in the SNI group and would be a confounding factor in our experiments. We aimed at having only one distinct variable that is the sham or the SNI surgery

We add that, all along the years, all animals with SNI in our lab (and in the lab of many colleagues using the SNI model), display nocifensive behavior to non-nociceptive stimuli (Von Frey threshold assays with up and down method) or lower latency to heat-induced pain-like behavior. We are thus entirely confident that the mice used in the present work would have pain-hypersensitivity like behavior, an indirect sign of peripheral neuropathic pain-like behavior.

Regarding the presence of spontaneous pain. We agree that this is an indirect sign of pain, which does not correspond to spontaneous pain, but the discussion on this topic is not in the scope of our work on sleep. We can only mention that other laboratories studied the appearance of signs of spontaneous pain in this model (Langford et al., 2010, doi: 10.1038/nmeth.1455; Tappe-Theodor and Kuner, 2014, doi: 10.1111/ejn.1264). To be very clear, we have now modified the text and use appropriate phrasing to avoid any confusion between an experimental model of neuropathic pain and the presence of spontaneous pain. As stated by the reviewers’ using experimental model of nerve injury and claiming that the animal is suffering from neuropathic pain is over conclusive, and it is the limitation that all pain scientists encounter. Nonetheless, the physiological data provided in the new Figure 2 provide new indirect signs – as said by the author – a signature of pain in the mice.

8. Authors cite several studies showing abnormal findings in α, β and low-γ during sleep (p4). These findings seem inconsistent with this paper. Please discuss (also make sure you're making apples-to-apples comparisons, including epoch lengths, normalization, model, etc.)

These portions of the introduction have been removed and the introduction rewritten to better comply with the specific points address in this study. Moreover, analysis for the frequency bands has been done and added in a new Figure 2, as discussed repeatedly above.

9. Figure legends too crowded with statistics.

We have now added a new Supplementary Table for Statistics and removed all detailed statistics from the figure legends.

10. Some icons in legends hard to interpret (e.g. Figure 1G).

We have now removed the small icons that referred to statistics. There are no longer icons that are hard to interpret on any figure.

11. What are units in Figure 1F?

These units represent power relative to baseline. The indication is present below the plots. The new figure arrangement presents this in a much clearer way.